# CLoRA: A Contrastive Approach to Compose Multiple LoRA Models

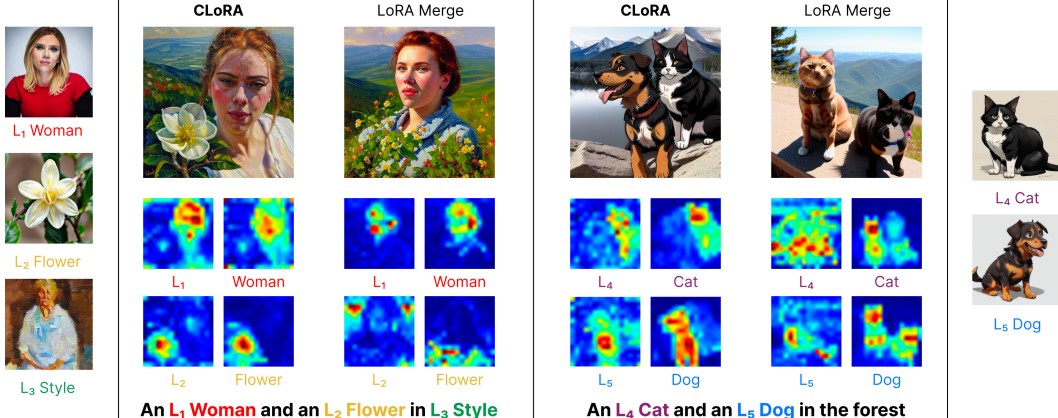

Figure 1: CLoRA is a training-free method that works on test-time, and uses contrastive learning to compose multiple concept and style LoRAs simultaneously. Using pre-trained LoRA models, such as $L_1$ for a person, and $L_2$ for a specific type of flower, the goal is to create an image that accurately represents both concepts described by their respective LoRAs. **Observation:** directly combining these LoRA models to compose the image often leads to poor outcomes (see LoRA Merge). This failure primarily arises because the attention mechanism fails to create coherent attention maps for subjects and their corresponding attributes. CLoRA revises the attention maps in test-time to clearly separate the attentions associated with distinct concept LoRAs.

## ABSTRACT

Low-Rank Adaptation (LoRA) has emerged as a powerful and popular technique for personalization, enabling efficient adaptation of pre-trained image generation models for specific tasks without comprehensive retraining. While employing individual pre-trained LoRA models excels at representing single concepts, such as those representing a specific dog or a cat, utilizing multiple LoRA models to capture a variety of concepts in a single image still poses a significant challenge. Existing methods often fall short, primarily because the attention mechanisms within different LoRA models overlap, leading to scenarios where one concept may be completely ignored (e.g., omitting the dog) or where concepts are incorrectly combined (e.g., producing an image of two cats instead of one cat and one dog). We introduce CLoRA, a training-free approach that addresses these limitations by updating the attention maps of multiple LoRA models at test-time, and leveraging the attention maps to create semantic masks for fusing latent representations. This enables the generation of composite images that accurately reflect the characteristics of each LoRA. Our comprehensive qualitative and quantitative evaluations demonstrate that CLoRA significantly outperforms existing methods in multi-concept image generation using LoRAs. Furthermore, we share our source code and benchmark dataset to promote further research.

# 1 INTRODUCTION

Diffusion text-to-image models (Ho et al., 2020) have revolutionized the generation of images from textual prompts, as evidenced by significant developments in models such as Stable Diffusion (Rombach et al., 2022), Imagen (Saharia et al., 2022), and DALL-E 2 (Ramesh et al., 2022). Their applications extend beyond image creation, including tasks like image editing (Avrahami et al., 2022b;a; Couairon et al., 2022; Hertz et al., 2022), inpainting (Lugmayr et al., 2022), and object detection (Chen et al., 2023). As generative models gaining popularity, personalized image generation plays a crucial role in creating high-quality, diverse images tailored to user preferences. Low-Rank Adaptation (Hu et al., 2021), initially introduced for LLMs, has emerged as a powerful technique for model personalization in image generation. LoRA models can efficiently fine-tune pre-trained diffusion models without the need for extensive retraining or significant computational resources. They are designed to optimize low-rank, factorized weight matrices specifically for the attention layers and are typically used in conjunction with personalization methods like DreamBooth (Ruiz et al., 2023). Since their introduction, LoRA models have gained significant popularity among researchers, developers, and artists (Gandikota et al., 2023; Guo et al., 2023). For example, Civit.ai[1], a widely used platform for sharing pre-trained models, hosts more than 100K LoRA models (Luo et al., 2024) tailored to specific characters, clothing styles, or other visual elements, allowing users to personalize their image creation experiences.

While existing LoRA models function as plug-and-play adapters for pre-trained models, integrating multiple LoRAs to facilitate the joint composition of concepts is an increasingly popular task. The ability to blend a diverse set of elements, such as various artistic styles or the incorporation of unique objects and people, into a cohesive visual narrative is crucial for leveraging compositionality (Huang et al., 2023b; Zhang et al., 2023). For example, consider a scenario where a user has two pre-trained LoRA models, representing a cat and a dog in a specific style (see Fig. 1). The objective might be to use these models to generate images of this particular cat and dog against various backgrounds or in different scenarios. However, using multiple LoRA models to create new, composite images has proven to be challenging, often leading to unsatisfactory results (see Fig. 1).

Prior works on combining LoRA models, such as the application of weighted linear combination of multiple LoRAs (Ryu, 2023), often lead to unsatisfactory outcomes where one of the LoRA concepts is often ignored. Other approaches (Shah et al., 2023; Huang et al., 2023a) train coefficient matrices to merge multiple LoRA models into a new one. However, these methods are limited by the capacity to merge only a single content and style LoRA (Shah et al., 2023) or by performance issues that destabilize the merging process as the number of LoRAs increases (Huang et al., 2023a). Other methods, such as Mix-of-Show (Gu et al., 2023), necessitate training specific LoRA variants such as Embedding-Decomposed LoRAs (EDLoRAs), diverging from the traditional LoRA models (e.g., civit.ai) commonly used within the community. They also depend on controls like regions defined by ControlNet (Zhang & Agrawala, 2023) conditions, which restrict their capability for condition-free generation. More recent works, such as OMG (Kong et al., 2024) utilizes off-the-shelf segmentation methods to isolate subjects during generation, with the overall effectiveness significantly dependent on the accuracy of the underlying segmentation model.

Contrary to these methods, we propose a solution that composes multiple LoRAs at test-time, without the need for training new models or specifying controls. Our approach involves adjusting the attention maps through latent updates during test-time to effectively guide the appropriate LoRA model to the correct area of the image while keeping LoRA weights intact. Our approach is inspired by the following novel observation: issues of 'attention overlap' and 'attribute binding', previously noted in image generation (Chefer et al., 2023; Agarwal et al., 2023), also exist in LoRA models. Attention overlap occurs when specialized LoRA models redundantly focus on similar features or areas within an image. This situation can lead to a dominance issue, where one LoRA model might overpower the contributions of others, skewing the generation process towards its specific attributes or style at the expense of a balanced representation (see Fig. 1). Another related issue is attribute binding, especially occurs in scenarios involving multiple content-specific LoRAs where features intended to represent different subjects blend indistinctly, failing to maintain the integrity and recognizability of each concept. For instance, consider the text prompt 'An $L_4$ cat and an $L_5$ dog in the forest' in Fig. 1, which depicts two LoRA models tailored for a specific cat and a dog, respectively. The

---

[1] http://civit.ai

straightforward approach of composing these LoRA models by merging the LoRA weights (see Fig. 1 -`LoRA Merge`) struggles to produce the intended results. This is because the $L_4$ attention, which should focus on the cat, blended with the $L_5$ attention, designated for the dog. Therefore, the output incorrectly features two cats, entirely omitting the dog. In contrast, our approach effectively refines the attention maps of the LoRA models in test-time to concentrate on the intended attributes, and produces an image that accurately places both LoRA models in their correct positions (see Fig. 1). Our framework, `CLoRA`, effectively composes multiple LoRA models while addressing the critical challenges of attention overlap and attribute binding. Our key contributions are as follows:

- We present a novel approach based on a contrastive objective to seamlessly integrate multiple content and style LoRAs simultaneously. Our approach works in test-time and does not require training.

- To the best of our knowledge, this work represents the first comprehensive attempt to observe and address attention overlap and attribute binding specifically within LoRA-enhanced image generation models. To address these issues, our method dynamically updates latents based on attention maps at test-time and fuses multiple latents using masks derived from cross-attention maps corresponding to distinct LoRA models.

- Unlike some of the previous methods, our approach does not need specialized LoRA variants and can directly use community LoRAs on civit.ai in a plug-and-play manner.

- We introduce a collection of LoRA models and prompts for multi-LoRA compositions, covering various characters, objects, and scenes. This collection establishes a standardized framework for evaluating the seamless integration of multiple concepts and style adaptations in LoRA-based image generation.

## 2 RELATED WORK

**Attention-based Methods for Improved Fidelity.** Text-to-image diffusion models often struggle with fidelity to input prompts, particularly when dealing with complex prompts containing multiple concepts or attributes (Tang et al., 2022). Recent advancements in high-fidelity text-to-image diffusion models (Chefer et al., 2023; Li et al., 2023; Agarwal et al., 2023) share our approach of utilizing attention maps to enhance image generation fidelity. A-Star (Agarwal et al., 2023) and DenseDiffusion (Kim et al., 2023) refine attention during the image generation process. Chefer et al. (2023) address neglected tokens in prompts, while Li et al. (2023) propose separate objective functions for missing objects and incorrect attribute binding issues. (Xie et al., 2023) and (Phung et al., 2024) utilize bounding boxes additional constraint to limit the generation of multiple subjects in constrained areas. While these methods tackle attention overlap and attribute binding within a single diffusion model, our approach uniquely addresses these issues across multiple LoRA models. Meral et al. (2023) use a contrastive approach on a single diffusion model, whereas our technique resolves these challenges across multiple diffusion models (LoRAs), each fine-tuned for distinct objects or attributes.

**Personalized Image Generation.** The field of personalized image generation has evolved significantly, building upon a rich history of image-based style transfer (Efros & Freeman, 2023; Hertzmann et al., 2023). Early advancements came through convolutional neural networks (Gatys et al., 2016; Huang & Belongie, 2017; Johnson et al., 2016) and GAN-based approaches (Karras et al., 2019; 2020; Chong & Forsyth, 2022; Gal et al., 2022b; Kwon & Ye, 2023). More recently, diffusion models (Ho et al., 2020; Rombach et al., 2022; Song et al., 2020) have offered superior quality and text control. In the context of large text-to-image diffusion models, personalization techniques have taken various forms. Textual Inversion (Gal et al., 2022a) and DreamBooth (Ruiz et al., 2023) focus on learning specific subject representations. LoRA (Ryu, 2023) and StyleDrop (Sohn et al., 2023) optimize for style personalization. Custom Diffusion (Kumari et al., 2023) attempts multi-concept learning but faces challenges in joint training and style disentanglement. (Zhang et al., 2024) uses attention calibration to disentangle multiple concepts from a single image and utilizes these concepts to generate personalized images.

**Merging Multiple LoRA Models.** The combination of LoRAs for simultaneous style and subject control is an emerging area of research, presenting unique challenges and opportunities. Existing approaches have explored various methods, each with its own limitations. Weighted summation, as

proposed by Ryu (2023), often yields suboptimal results due to its simplicity. Gu et al. (2023) suggest retraining specific EDLoRA models for each concept, which limits the approach's applicability to existing community LoRAs. Wu et al. (2023) propose composing LoRAs through a mixture of experts, but this method requires learnable gating functions that must be trained for each domain. Test-time LoRA composition methods, such as Multi LoRA Composite and Switch by Zhong et al. (2024), have also been proposed, but these do not operate on attention maps and may produce unsatisfactory results. ZipLoRA (Shah et al., 2023) synthesizes a new LoRA model based on a style and a content LoRA, however their method falls short in handling multiple content LoRAs. OMG by Kong et al. (2024) utilizes off-the-shelf segmentation methods to isolate subjects during generation, with its performance heavily dependent on the multi-object generation fidelity of diffusion models and the accuracy of the underlying segmentation model. (Yang et al., 2024) ==proposes a training-free approach tackling concept confusion by introducing additional injection and isolation constraints using user-provided bounding boxes.== Our approach distinguishes itself by directly addressing attention overlap and attribute binding issues in the context of multiple LoRA models. We incorporate test-time generated masks, enhancing the disentanglement of LoRA models and effectively resolving attention map and attribute binding problems. This offers a more comprehensive solution for high-fidelity, multi-concept image generation, bridging the gap between single-model attention refinement and effective LoRA model composition.

## 3 METHODOLOGY

This section outlines the foundational concepts of diffusion models, and Low-Rank Adaptation, followed by a detailed discussion of our novel approach, CLoRA (see Fig. 2).

### 3.1 BACKGROUND

**Diffusion models.** Our method is implemented on the Stable Diffusion 1.5 (SDv1.5) model, a state-of-the-art text-to-image generation framework for LoRA applications. Stable Diffusion operates in the latent space of an autoencoder, comprising an encoder $\mathcal{E}$ and a decoder $\mathcal{D}$. The encoder maps an input image $x$ to a lower-dimensional latent code $z = \mathcal{E}(x)$, while the decoder reconstructs the image from this latent representation, such that $\mathcal{D}(z) \approx x$. The core of Stable Diffusion is a diffusion model (Ho et al., 2020) trained within this latent space. The diffusion process gradually adds noise to the original latent code $z_0$, producing $z_t$ at timestep $t$. A UNet-based (Ronneberger et al., 2015) denoiser $\epsilon_\theta$ is trained to predict and remove the noise. The training objective is defined as:

$$\mathcal{L} = \mathbb{E}_{z_t, \epsilon \sim \mathrm{N}(0,\mathrm{I}), c(\mathcal{P}), t} \left[ \| \epsilon - \epsilon_\theta(z_t, c(\mathcal{P}), t) \|^2 \right] \tag{1}$$

where $c(\mathcal{P})$ represents the conditional information derived from the text prompt $\mathcal{P}$. Stable Diffusion employs CLIP (Radford et al., 2021) to embed the text prompt into a sequence $c$, then fed into the UNet through cross-attention mechanisms. In these layers, $c$ is linearly projected into keys ($K$) and values ($V$), while the UNet's intermediate representation is projected into queries ($Q$). The attention at time $t$ is then calculated as $A_t = \mathrm{Softmax}(QK^{\intercal}/\sqrt{d})$. These attention maps $A_t$ can be reshaped into $\mathbb{R}^{h \times w \times l}$, where $h$ and $w$ are the height and width of the feature map (typically $16 \times 16$, $32 \times 32$, or $64 \times 64$), and $l$ is the text embedding sequence length. Our work utilizes the $16 \times 16$ attention maps, which capture the most semantically meaningful information (Hertz et al., 2022).

**LoRA models.** LoRA fine-tunes large models by introducing rank-decomposition matrices while freezing the base layer. In SD fine-tuning, LoRA is applied to cross-attention layers responsible for text and image connection. Formally, a LoRA model is represented as a low-rank matrix pair ($W_{\mathrm{out}}$, $W_{\mathrm{in}}$). These matrices capture the adjustments introduced to the $W$ weights of a pre-trained model ($\theta$). The updated weights during image generation are calculated as $W' = W + W_{in}W_{out}$. The low-rank property ensures that ($W_{\mathrm{out}}$ and $W_{\mathrm{in}}$) have significantly smaller dimensions compared to full-weight matrices, resulting in a drastically reduced file size for the LoRA model. For example, while a full SDv1.5 model is about 3.44GB, a LoRA model typically ranges from 15 to 100 MB.

**Contrastive learning.** Contrastive learning has emerged as a powerful method in representation learning (Chen et al., 2020; Oord et al., 2018). Its core principle is bringing similar data points closer together in a latent embedding space while pushing dissimilar ones apart. Let $x \in \mathcal{X}$ represent an input data point, with $x^+$ denoting a positive pair (both $x$ and $x^+$ share the same label) and $x^-$

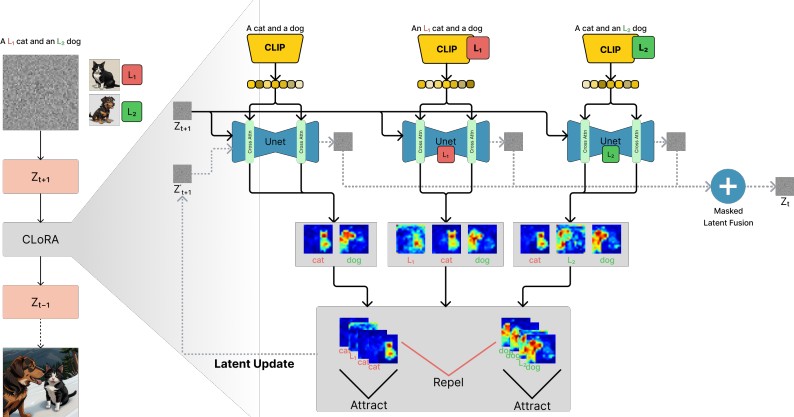

Figure 2: Overview of CLoRA, a training-free, test-time approach for composing multiple LoRA models. Our method accepts a user-provided text prompt, such as 'An $L_1$ cat and an $L_2$ dog,' along with their corresponding LoRA models $L_1$ and $L_2$. CLoRA applies test-time optimization to attention maps to address attention overlap and attribute binding issues using a contrastive objective.

denoting a negative pair (where the data points have different labels). The function $f : \mathcal{X} \rightarrow \mathbb{R}^N$ is an encoder that maps an input $x$ to an N-dimensional embedding vector. Various contrastive learning objectives are proposed such as InfoNCE (also known as NT-Xent) (Oord et al., 2018) which we utilize in this work.

## 3.2 CLoRA

Given a text prompt such as 'An $L_1$ cat and an $L_2$ dog,' and their corresponding LoRA models $L_1$ and $L_2$, our method aims to create an image that reflects the text prompt while respecting the corresponding LoRA models (see Fig. 2). Our method refines the attention maps of the LoRA models at test-time using a contrastive objective. This objective encourages the attention maps to focus on the intended attributes, thereby resolving issues of attention overlap and attribute binding. Next, we discuss the key components of our contrastive objective and explain how positive and negative pairs are formed.

For simplicity, let us assume that we have two LoRA models to compose. Note that for ease of understanding the positive pairs will be shown in the same color coding such as $L_1\ S_1$ and $L_2\ S_2$. First, we decompose the user-provided prompt into components that align with specific concepts ($S_1$ and $S_2$), defined by different LoRAs ($L_1$ and $L_2$). For example, given the prompt 'an $L_1\ S_1$ and an $L_2\ S_2$' (*e.g.,* 'An $L_1$ cat and an $L_2$ dog,'), where the LoRA models $L_1$ and $L_2$ represent the personalized concepts for $S_1$ and $S_2$, respectively, we employ three prompt variations. First is the original prompt, 'an $S_1$ and an $S_2$'. Second is the $L_1$-applied prompt, 'an $L_1\ S_1$ and an $S_2$'. Lastly, $L_2$-applied prompt, 'an $S_1$ and an $L_2\ S_2$'. We then generate corresponding text embeddings using the CLIP model. If the text encoder was fine-tuned during LoRA training, the embeddings are generated using the fine-tuned text encoder. Otherwise, we use the embeddings from the base model. These prompt variations will be used to form positive and negative pairs during the contrastive objective.

During the image generation process, Stable Diffusion utilizes cross-attention maps to guide attention on specific image regions at each diffusion step. However, as discussed before, these attention maps suffer from attention overlap and attribute binding issues, leading to unsatisfactory compositions. We apply a test-time optimization to the attention maps to encourage that each concept (e.g., '$S_1$' for the cat or '$S_2$' for the dog) is represented according to their corresponding LoRA. In order to do this, we first categorize cross-attention maps based on their corresponding tokens in the prompts, creating concept groups, $C_1$ and $C_2$. For the first group, $C_1$, we include the cross-attention map for $S_1$ from the original prompt, cross-attention maps for $L_1$ and $S_1$ from the $L_1$-applied prompt, and the cross-attention map for $S_1$ from the $L_2$-applied prompt. Similarly, for the second group, $C_2$, we include the cross-attention map for $S_2$ from the original prompt, the cross-attention map for $S_2$ from the $L_1$-applied prompt, and cross-attention maps for $L_2$ and $S_2$ from the $L_2$-applied prompt. This grouping will be utilized in our contrastive objective to ensure that the diffusion process maintains a

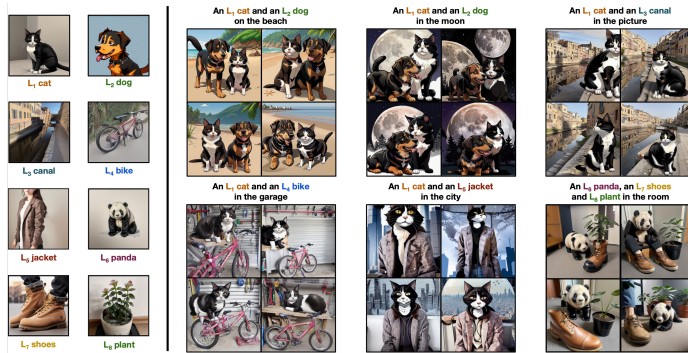

Figure 3: **The qualitative results produced by CLoRA** showcase a range of compositions, including animal-animal, object-object, and animal-object pairs. Left columns display sample images generated by the individual LoRA models. Our approach is successful at composing multiple content LoRAs—for example, combining a *cat* and a *dog*—along with *scene* LoRAs, such as pairing a *cat* with a *canal* scene. Moreover, it demonstrates the capability to integrate more than two LoRAs, exemplified by the composition of a *panda*, *shoe*, and *plant* LoRA (see bottom right).

coherent understanding of each concept while integrating the stylistic variations introduced by the LoRAs. Separating these concepts will also prevent attention overlap between different concepts, ensuring that each element of the prompt is faithfully represented in the generated image.

**CLoRA Contrastive Objective:** We design a contrastive objective during inference to maintain consistency with the input prompt. We used the form of InfoNCE loss due to its fast convergence (Oord et al., 2018). Our loss function takes pairs of cross-attention maps, processing pairs within the same group as positive and pairs from different groups as negative. For example, given the text prompt 'An $L_1$ cat and an $L_2$ dog,' and their corresponding concept groups $C_1$ ('cat' and $L_1$) and $C_2$ ('dog' and $L_2$), the attention maps of the concept group $C_1$ form positive pairs. In other words we want the attention map for the cat from the original prompt and the attention map for $L_1$ from the $L_1$-applied prompt get close to each other since we want $L_1$ LoRA to be aligned with its corresponding subject, cat. In contrast, the attention maps of different concept groups $C_1$ and $C_2$ (e.g., the attention map for cat and dog from the original prompt) form negative pairs since we want these attention maps to repel each other to avoid attention overlap issue (see Fig. 2 for an illustration). The loss function for a single positive pair is expressed as:

$$\mathcal{L} = -\log \frac{\exp(\text{sim}(A^j, A^{j^+})/\tau)}{\sum_{n \in \{j^+, j_1^-, \cdots j_N^-\}} \exp(\text{sim}(A^j, A^n)/\tau)} \tag{2}$$

where cosine similarity $\text{sim}(u, v)$ is defined as $\text{sim}(u, v) = u^T \cdot v / \|u\| \|v\|$. Here, $\tau$ is the temperature parameter, and the denominator includes one positive pair and all negative pairs for $A^j$. $N$ is the number of negative pairs that include $A^j$. The overall InfoNCE loss is averaged across all positive pairs.

**Latent Optimization.** The loss function guides the latent representation during the diffusion process. The latent representation is updated iteratively similar to Chefer et al. (2023) and Agarwal et al. (2023): $z_t' = z_t - \alpha_t \nabla_{z_t} \mathcal{L}$ where $\alpha_t$ is the learning rate at step $t$.

**Masked Latent Fusion.** In our approach, after a backward step in the diffusion process, we combine the latent representations generated by Stable Diffusion with those derived from additional LoRA models. While the direct combination of these latents is possible as described by Bar-Tal et al. (2023), we introduce a masking mechanism to ensure that each LoRA influences only the relevant regions of the image. This is achieved by leveraging attention maps from the corresponding LoRA outputs to create binary masks. To create the masks, we first extract attention maps for the relevant tokens from each LoRA-applied prompt. For $L_1$, we use the attention maps corresponding to the tokens $L_1$ and $S_1$ from the $L_1$-applied prompt, 'an $L_1$ $S_1$ and an $S_2$'. Similarly, for $L_2$, we extract the attention maps for the tokens $L_2$ and $S_2$ from the $L_2$-applied prompt, 'an $S_1$ and an $L_2$ $S_2$'. To create binary masks, we apply a thresholding operation to these attention maps, following a method akin to semantic

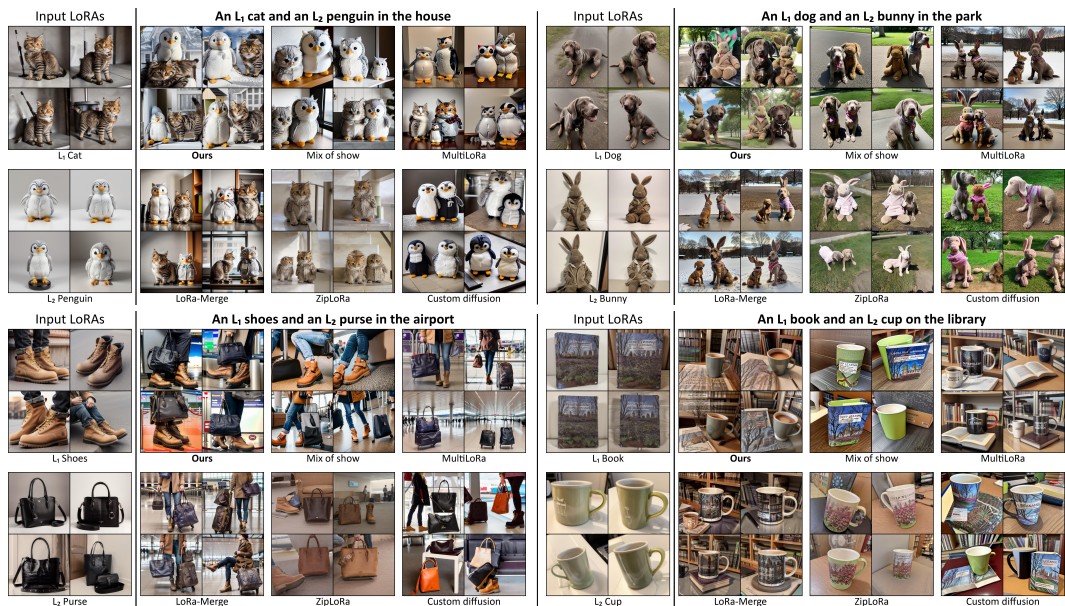

Figure 4: **Qualitative Comparison** of CLoRA, Mix of Show, MultiLoRA, LoRA-Merge, ZipLoRA and Custom Diffusion. Our method can generate compositions that faithfully represent the LoRA concepts, whereas other methods often overlook one of the LoRAs and generate a single LoRA concept for both subjects. Please zoom-in for more details. See Appendix for more comparisons.

segmentation described by Tang et al. (2022). For each position $(x, y)$ in the attention map, the binary mask value $M[x, y]$ is determined using the equation $M[x, y] = \mathbb{I}(A[x, y] \geq \lambda \max_{i,j} A[i, j])$ where $M[x, y]$ represents the binary mask output, $A[x, y]$ is the attention map value at position $(x, y)$ for the corresponding token, $\mathbb{I}(\cdot)$ is the indicator function that outputs 1 if the condition is true (and 0 otherwise), and $\lambda$ is a threshold value between 0 and 1. This thresholding process ensures that only areas with attention values exceeding a certain percentage of the maximum attention value in the map are included in the mask. When multiple tokens contribute to a single LoRA (such as '$L_1$' and '$S_1$' for $L_1$), we perform a union operation on the individual masks to ensure that any region receiving attention from either token is included in the final mask for that LoRA. This masking procedure restricts the influence of each LoRA to the relevant regions, thereby preserving the integrity of the generated image while incorporating the specific stylistic elements defined by the LoRAs.

## 4 EXPERIMENTS

In this section, we present qualitative results, along with quantitative comparisons and a user study. For additional results, please refer to our supplementary material.

**Datasets.** Due to the absence of standardized benchmarks for composing multiple LoRA models, we compile a set of 131 LoRA models. These models include custom characters generated with the character sheet trick (see Appendix D) and various concepts from Custom Concept dataset (Kumari et al., 2023). These models are accompanied by 200 prompts, such as 'A plushie bunny and a flower in the forest,' where both 'plushie bunny' and 'flower' have corresponding LoRA models. Additional details about the dataset and composition prompts can be found in the Appendix D.

**Implementation Details.** For each prompt, we use 10 different seeds, running 50 iterations with Stable Diffusion v1.5. Following Chefer et al. (2023), we apply optimization in iterations $i \in \{0, 10, 20\}$, and stop further optimization after $i = 25$ to prevent artifacts. For contrastive learning, we set the temperature to $\tau = 0.5$ in Equation 2. Image generation was performed on a V100 GPU. Our approach takes $\approx 25$ seconds to compose two LoRA models, and can successfully combine up to eight LoRAs on a single H100 Nvidia GPU. See Appendix A for more details.

**Baselines.** We compare our results with baselines such as LoRA-Merge (Ryu, 2023) that merges LoRAs as a weighted combination, ZipLoRA (Shah et al., 2023) that synthesizes a new LoRA model based on the provided LoRAs, Mix-of-Show (Gu et al., 2023) that requires training a specific LoRA

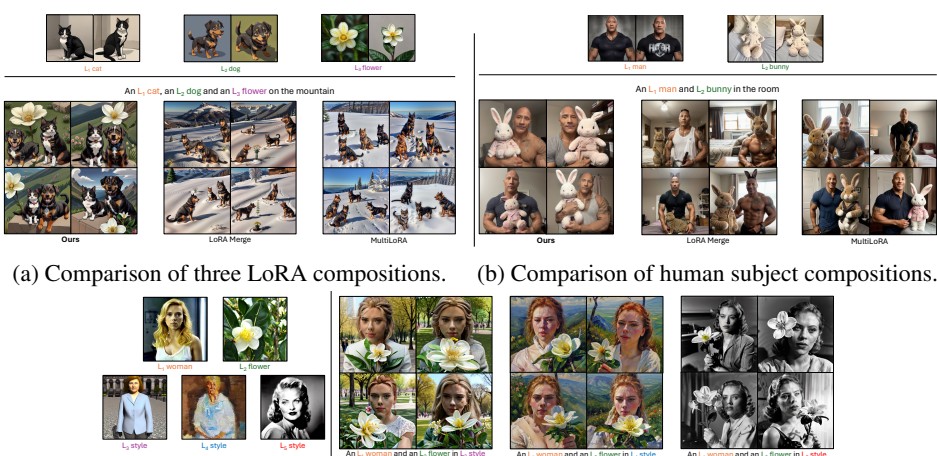

(a) Comparison of three LoRA compositions.     (b) Comparison of human subject compositions.

(c) Results showcasing the composition of two subject and one style LoRAs.

Figure 5: **Qualitative Results and Comparisons of CLoRA.** (a) Our method can successfully compose images using three LoRAs. (b) Our method can handle realistic compositions featuring humans. (c) Our method can seamlessly compose images using style, object, and human LoRAs.

type, Custom Diffusion (Kumari et al., 2023) and MultiLoRA (Zhong et al., 2024). For MultiLoRA, we use the 'Composite' configuration, as it outperformed MultiLoRA-Switch (Zhong et al., 2024).

## 4.1 QUALITATIVE EXPERIMENTS

**Qualitative Results.** The qualitative performance of our approach is shown in Fig. 1 and 3. Our method successfully composes images using multiple content LoRAs, such as a *cat* and *dog*, within varied backgrounds like the *mountain* or *moon* (Figs. 1 and 3). Furthermore, it successfully composes a content LoRA with a scene LoRA, such as situating the *cat* within a specific *canal* as defined by the scene LoRA (Fig. 3). Our method also demonstrates versatility, combining diverse LoRAs, such as pairing a *cat* with a *bicycle* or *clothing* (Fig. 3). Notably, it handles compositions involving more than two LoRAs, as illustrated by a *panda*, *shoe*, and *plant* in the bottom right of Fig. 3.

**Qualitative Comparison** We provide a qualitative comparison between our method and several baselines in Fig. 4 , focusing on animal-animal and object-object compositions. Each comparison visualizes four randomly generated compositions using our method, Mix of Show (Gu et al., 2023), MultiLoRA (Zhong et al., 2024), LoRA-Merge (Ryu, 2023), ZipLoRA (Shah et al., 2023), and Custom Diffusion (Kumari et al., 2023). Our method faithfully captures both concepts from the corresponding LoRA models without attention overlap issues. Other approaches often struggle with attribute binding or fail to represent one of the concepts due to overlapping attention maps. For example, in a prompt such as 'An $L_1$ cat and an $L_2$ penguin in the house' (where $L_1$ represents a cat LoRA and $L_2$ a plush penguin LoRA), Mix of Show blends the two objects, producing either two plush penguins while ignoring the *cat*, or a single *cat* with plush-like features (Fig. 4, top-left). MultiLoRA fails to resemble the specific LoRA models, producing either two *cats* or two *penguins*. LoRA-Merge generates a *cat* that somewhat aligns with the intended LoRA but does not capture the *penguin* accurately. ZipLoRA frequently fails to incorporate the plush *penguin*, instead creating two *cats* due to its design constraints for combining multiple content LoRAs. Similarly, Custom Diffusion often overlooks the *cat* LoRA, focusing only on generating the plush *penguin*. Similar observations can be made when combining object-object LoRAs (see Fig. 4 bottom row). Our method successfully generates both elements within a composition, e.g. effectively positioning a specific pair of shoes and a purse as dictated by different LoRA models (Fig. 4, bottom-left). In contrast, other approaches frequently miss one of the elements or create objects that do not match the characteristics outlined by the respective LoRAs. Additionally, these methods often struggle with attribute binding issues. This problem is evident in Fig. 4 (bottom right), where the book LoRA tends to blend with the cup LoRA, leading to an image of a cup that features the cover of the book. We also note that our method struggles to depict the identity of the book and the cup objects, however it is still able to create a composition without blending the objects. Please see Appendix G for additional comparisons.

Table 1: Average, Minimum/Maximum DINO image-image similarities, and CLIP-I and CLIP-T metrics between the merged prompts and individual LoRA models, User Study. For all metrics, the higher, the better.

| | | Merge Ryu (2023) | Composite | Switch Zhong et al. (2024) | ZipLoRA Shah et al. (2023) | Mix-of-Show Gu et al. (2023) | Ours |
|---|---|---|---|---|---|---|---|
| DINO | Min. | 0.376 ± 0.041 | 0.288 ± 0.049 | 0.307 ± 0.055 | 0.369 ± 0.036 | 0.407 ± 0.035 | **0.447 ± 0.035** |
| | Avg. | 0.472 ± 0.036 | 0.379 ± 0.045 | 0.395 ± 0.053 | 0.496 ± 0.030 | 0.526 ± 0.024 | **0.554 ± 0.028** |
| | Max. | 0.504 ± 0.038 | 0.417 ± 0.046 | 0.432 ± 0.055 | 0.533 ± 0.032 | 0.564 ± 0.024 | **0.593 ± 0.024** |
| CLIP-I | Min | 0.641 ± 0.029 | 0.614 ± 0.035 | 0.619 ± 0.039 | 0.659 ± 0.022 | 0.664 ± 0.023 | **0.683 ± 0.017** |
| | Avg | 0.683 ± 0.029 | 0.654 ± 0.035 | 0.659 ± 0.036 | 0.707 ± 0.021 | 0.712 ± 0.022 | **0.725 ± 0.017** |
| | Max | 0.714 ± 0.028 | 0.690 ± 0.033 | 0.695 ± 0.036 | 0.740 ± 0.021 | 0.744 ± 0.023 | **0.756 ± 0.017** |
| CLIP-T | | 0.814 ± 0.054 | 0.833 ± 0.091 | 0.822 ± 0.089 | 0.767 ± 0.081 | 0.760 ± 0.074 | **0.862 ± 0.052** |
| User Study | | 2.0 ± 1.10 | 2.11 ± 1.12 | 1.98 ± 1.14 | 2.81 ± 1.18 | 2.03 ± 1.12 | **3.32 ± 1.13** |

**Composition with three LoRA models.** We evaluate the ability to compose with more than two LoRA models in Fig. 5a. Our method effectively maintains the characteristics of each LoRA in the composite image, while other methods struggle to create coherent compositions, often blending multiple models together[2]. Moreover, Fig. 5c shows sample compositions using 3 LoRAs that corresponds to style, object and human LoRAs.

**Composition with human subjects.** We compare the composition of human subjects in Figs. 1 and 5b. Our method seamlessly composes human subjects with objects, preserving the distinct properties of each LoRA. Other methods often struggle to integrate both elements effectively (see Fig. 5b).

**Composition with style LoRAs.** Our approach can blend both style and concept LoRAs (see Figs. 1 and 5c). The results show that our method captures the unique features of each content LoRA (e.g., a flower and a human), while applying the style LoRA consistently across the entire image.

## 4.2 QUANTITATIVE EXPERIMENTS

**Quantitative Comparison.** We leverage DINO and CLIP features (Radford et al., 2021) to assess the quality of images generated by our method and compare methods that combine multiple LoRAs. DINO offers a hierarchical representation of image content, enabling a more detailed analysis of how each LoRA contributes to specific aspects of the merged image. To calculate DINO-based metrics, we first generate separate outputs using each individual LoRA based on the prompt sub-components (e.g., $L_1$ cat' and $L_2$ flower'). Then, we extract DINO features for the merged image and each single LoRA output. Finally, we calculate cosine similarity between the DINO features of the merged image and the corresponding features from each single LoRA output.

We utilize three DINO-based metrics: *Average DINO Similarity*, which reflects the overall alignment between the merged image and individual LoRAs averaged across all LoRAs; *Minimum DINO Similarity*, which uses the cosine similarity between the DINO features of the merged image and the least similar LoRA reference output; and *Maximum DINO Similarity*, which identifies the LoRA reference image whose influence is most represented in the merged image. For each LoRA model and composition prompts, 50 reference images are generated and DINO similarities are calculated. Prompts used in benchmarks consist of two subjects and a background, such as 'an $L_1$ cat and an $L_2$ penguin in the house' (see Fig. 4). The results (see Table 1) demonstrate that our method surpasses the baselines in terms of faithfully merging content from LoRAs.

Additionally, we include comparisons using CLIP-I (image-to-image similarity) and CLIP-T (image-to-text similarity) metrics to evaluate the performance of our method against competing approaches (see Tab. 1). The results demonstrate that CloRA consistently outperforms other methods across both metrics, highlighting its ability to generate images that align with the intended concepts and prompts.

**User Study.** To further validate our approach, we conducted a user study involving 50 participants recruited through the Prolific platform[3]. Each participant was shown four generated images per composition from different methods and asked to rate how faithfully each method preserved the concepts represented by the LoRAs (on a scale from 1 = "Not faithful" to 5 = "Very faithful"). As presented in Table 1, our method consistently outperformed the baseline approaches, achieving higher scores for faithful representation of concepts.

---

[2]Some methods were excluded because they could not compose three LoRAs (Shah et al., 2023), or require additional controls (Gu et al., 2023).

[3]http://prolific.com.

Figure 6: CLoRA Ablation Study. Using the $L_1$ cat and $L_2$ dog LoRAs, the effects of two key components (latent update and latent masking) can be observed.

**Ablation Study** Our method integrates two key components to generate compositions with multiple LoRAs: *Latent Update* and *Latent Masking*. *Latent Update* employs our contrastive objective to direct the model's attention precisely towards the concepts specified by each LoRA, preventing misdirection and attention to irrelevant areas. Without this component, the model could erroneously generate duplicate objects or incorrect attribute connections (e.g., producing two dogs instead of a cat and a dog), as shown in Fig. 6. *Latent Masking* protects the identity of the main subject during generation. Without masking, every pixel would be influenced by all prompts, leading to inconsistencies and loss of identity in the final image. Together, these components enhance composition process, enabling users to introduce specific styles or variations into designated regions guided by multiple LoRAs.

## 5 LIMITATIONS

Our method marks a significant advancement in creative fields, enabling users to create compositions from multiple LoRA models. However, while democratizing creativity, our method raises concerns about ethical implications of automated tools in art creation, necessitating thoughtful discourse around their use Kenthapadi et al. (2023). Additionally, the ease of generating personalized images could lead to misuse for malicious purposes, such as creating deepfakes or spreading misinformation, as highlighted by Korshunov & Marcel (2018). Additionally, integrating and optimizing multiple LoRA models simultaneously poses a challenge due to potential increases in computational complexity, which can affect processing times and resource demands as the number of LoRA models increases, a limitation that is also common among competing methods. Nevertheless, our method is capable of successfully combining up to four LoRAs on a single Nvidia H100 GPU, taking between 25 seconds (2 LoRAs) up to 90 seconds (8 LoRAs), while consuming a memory from 25GB to 80GB, respectively (see Fig. 7). A more detailed discussion is provided in App. A.

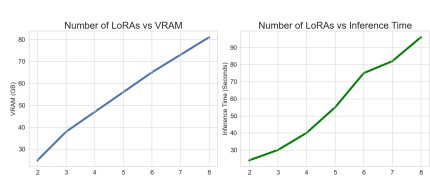

Figure 7: **Number of LoRAs vs. VRAM usage (left) and inference time (right).**

## 6 CONCLUSION

In this paper, we presented a training-free method, CLoRA, for integrating content from multiple LoRAs to compose images. Our approach addresses the limitations of existing methods by dynamically adjusting attention maps in test-time, ensuring each LoRA guides the diffusion process toward its designated subject. Furthermore, we provide a benchmark LoRA and composition prompt dataset for a thorough evaluation. Our experimental results demonstrate that CLoRA significantly outperforms existing baselines across various metrics, including DINO-based similarity, CLIP alignment, and user study evaluations, showcasing its robustness in faithfully representing and blending multiple LoRAs. Unlike competing methods, our approach does not require the training of specific LoRAs and is compatible with a wide range of community-developed LoRAs available on platforms like Civit.ai. By making our source code and LoRA collection publicly available, we aim to promote transparency and reproducibility, as well as encourage further advancements in this area. We envision CLoRA as a valuable tool for democratizing creativity in visual generative AI, enabling broader adoption and innovation in applications ranging from digital art and storytelling to gaming.

## 7 REPRODUCIBILITY STATEMENT

To promote reproducibility and facilitate further research, we have made our source code publicly available in the supplementary materials. Detailed descriptions of our experimental procedures are thoroughly outlined in the main paper under 'Implementation Details' in Section 4. Additionally, comprehensive information about our LoRA collection is provided in Appendix D.

We also offer an extensive collection of uncurated qualitative comparisons between our method and those of competitors, which can be found in Appendix G. This extensive compilation aims to provide a robust and comprehensive assessment of our approach compared to existing methods. For our quantitative analyses, we include standard deviations for all metrics, which are presented in Table 1 to ensure transparency and reliability of the reported results.

## 8 ETHICS STATEMENT

While our method democratizes creativity by simplifying the process of art creation, it also introduces ethical considerations that must be taken into account. Our method enable the generation of personalized images with minimal effort, and opens the door to transformative opportunities in art and design. However, as noted by Kenthapadi et al. (2023), it necessitates a comprehensive and thoughtful discourse around their ethical use to prevent potential abuses. In addition to these concerns, our user study strictly adheres to anonymity protocols to safeguard participant privacy.

The capability of our method to effortlessly generate personalized images also poses risks of misuse in several harmful ways, such as the creation of deepfakes. These can be used to forge identities or manipulate public opinion, a concern underscored by Korshunov & Marcel (2018).

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

# A RUNTIME PERFORMANCE AND IMPACT OF NUMBER OF LoRAs

## A.1 COMPARISON OF METHODS IN TERMS OF RUNTIME.

This section presents a comparison of various methods in terms of their compatibility with CivitAI (civ, 2020), VRAM requirements, and runtime performance. Table 2 summarizes the results. All experiments were conducted on an NVIDIA H100 GPU with 80GB of VRAM.

The methods evaluated include Custom Diffusion, LoRA Merge, Multi-LoRA (composite and switch modes), Mix-of-Show, ZipLoRA, OMG, LoRA-Composer and our proposed method. Methods like Custom Diffusion and Mix-of-Show are not compatible with CivitAI, while others, such as LoRA Merge and the proposed method, are fully compatible.

Our proposed method demonstrates a favorable balance between VRAM usage and runtime performance. It achieves faster inference times compared to methods like ZipLoRA and OMG, while maintaining a moderate VRAM requirement of 25GB. This makes it a practical choice for scalable and efficient multi-concept image generation tasks.

| Method | CivitAI Compatibility | VRAM (Finetuning/Inference) | Runtime (Finetuning/Inference) |
|---|---|---|---|
| Custom Diffusion | × | 28GB + 8GB | 4.2 min + 3.5s |
| LoRA Merge | ✓ | 7GB | 3.2s |
| Multi-LoRA - composite | ✓ | 7GB | 3.4s |
| Multi-LoRA - switch | ✓ | 7GB | 4.8s |
| Mix-of-Show | × | 10GB + 10GB | 10min + 3.3s |
| ZipLoRA | ✓ | 39GB + 17GB | 8min + 4.2s |
| OMG | ✓ | 30GB | 62s |
| LoRA-Composer | × | 51GB | 35s |
| Ours | ✓ | 25GB | 24s |

Table 2: Comparison of methods in terms of CivitAI compatibility, VRAM usage, and runtime.

As shown in Tab. 2, our proposed method outperforms many existing approaches in inference time while maintaining reasonable VRAM requirements. This makes it a practical choice for scalable and efficient deployments.

## A.2 EFFECT OF NUMBER OF LoRAs ON RUNTIME AND VRAM USAGE.

Figure 8 illustrates the relationship between the number of LoRAs and their impact on VRAM usage and inference runtime. As the number of LoRAs increases, both VRAM consumption and inference time show a gradual and predictable growth. For instance, moving from 2 LoRAs to 8 LoRAs results in an increase in VRAM usage from 25 GB to 81 GB and inference time from 24 seconds to 96 seconds. These trends indicate that while additional LoRAs enhance multi-concept flexibility, the associated computational requirements grow in a manageable and predictable manner, making them a practical choice for many applications. All results were obtained using NVIDIA H100 GPUs with 80GB VRAM.

# B USER STUDY DETAILS

We recruited 50 participants through the Prolific platform[4]. Each participant was shown 48 images, and asked to rate how faithfully each method preserved the concepts represented by the LoRAs (on a scale from 1 = "Not faithful" to 5 = "Very faithful"). The order of images were randomized per participant. Please see Fig. 9 to see a screenshot of our user study.

---

[4] http://prolific.com.

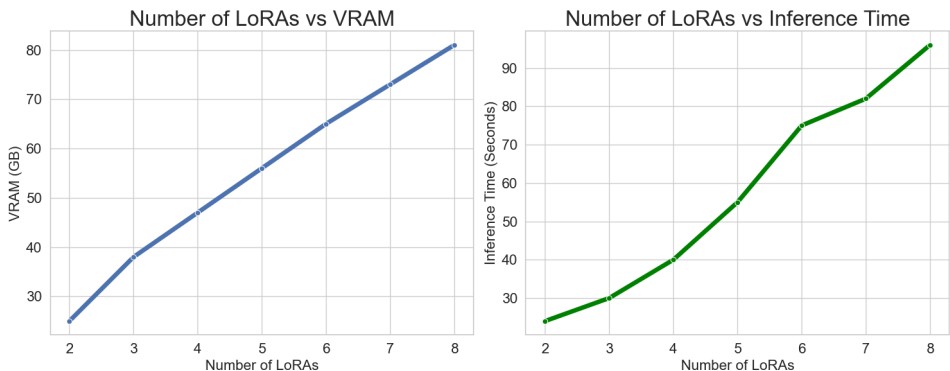

Figure 8: Number of LoRAs vs. VRAM usage (left) and inference time (right).

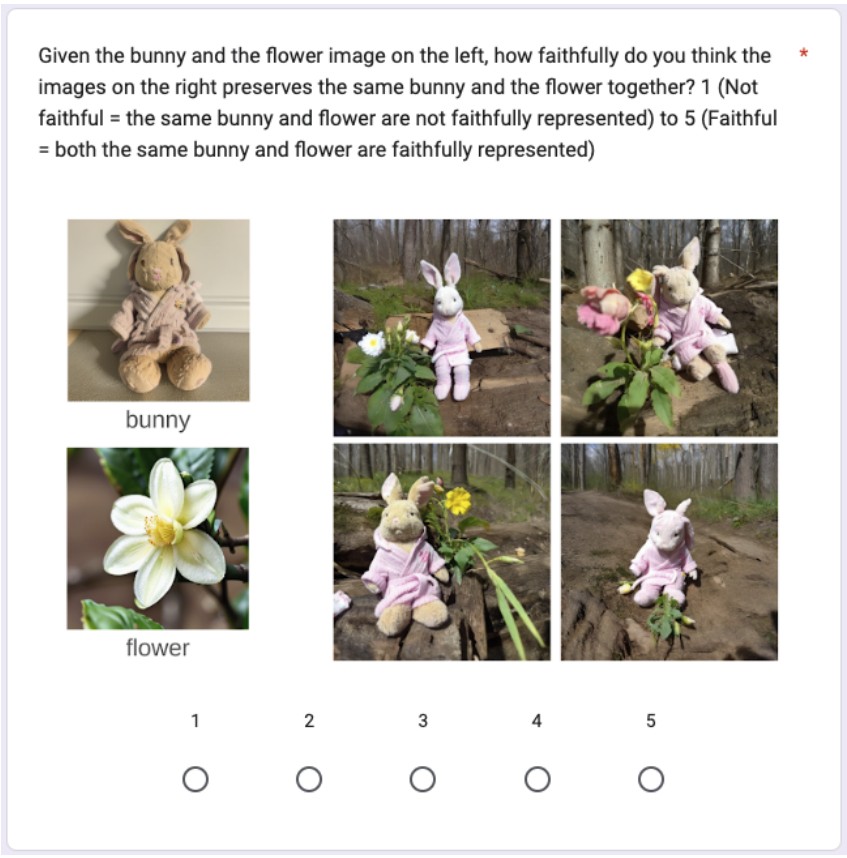

Figure 9: **Screenshot of our user study.** Each participant was shown images generated by LoRA models (on the left) and 4 images generated by the method (ours or competitors). Users are then requested to rate from 1-5 (Not faithful/Faithful) based on how well the generated images reflect the concepts depicted in the LoRA models.

## C ADDITIONAL RESULTS

Figure 10 shows CLoRA's capabilities of generating images with similar subjects. Figure 11 showcases the CLoRA's ability to merge LoRAs in complex and interacting scenes.

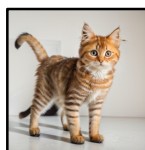
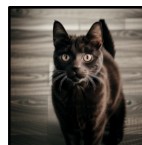
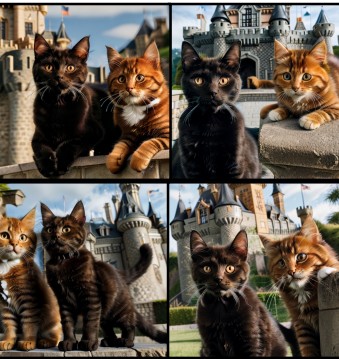
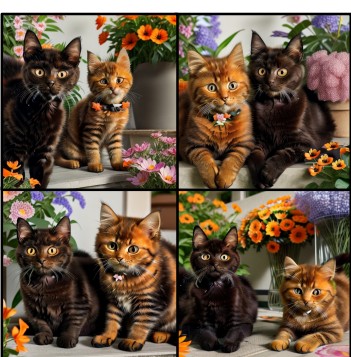

**L₁ cat**

**L₂ cat**

An L₁ cat and an L₂ cat
in front of a castle in a fantastic land

An L₁ cat and an L₂ cat
with vibrant flowers

Figure 10: Qualitative results showing that `CLoRA` is capable of generating images using LoRAs that has similar subjects.

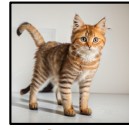
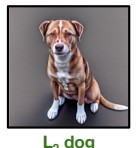
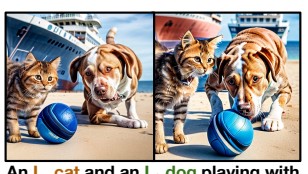
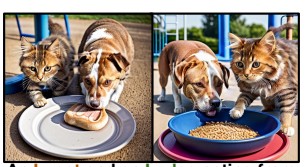

**L₁ cat**          **L₂ dog**

An L₁ cat and an L₂ dog playing with
a ball together, near the beach with a
ship in the background

An L₁ cat and an L₂ dog eating from
the same plate, in a playground

Figure 11: Qualitative Results showing that `CLoRA` is capable of composing images in complex interacting scenes.

# D  DETAILS OF BENCHMARK LORA COLLECTION

We propose 131 pre-trained LoRA models and 200 text-prompts for multi-LoRA composition. The details of our dataset is given below.

## D.1  DATASETS

This study leverages two key datasets for benchmark:

- **Custom collection:** We generated custom characters such as cartoon style *cat* and *dog*, created using the *character sheet* trick [5] popular within the Stable Diffusion community. This set comprises 20 unique characters, where we trained a LoRA per character.

- **CustomConcept101:** We used a popular dataset Kumari et al. (2023) CustomConcept101 that includes several diverse objects such as *plushie bunny*, *flower*, and *chair*. All 101 concepts are utilized.

Leveraging the datasets above, we trained LoRAs to represent each concept, totaling to 131 LoRA models. For every competitor, the base stable diffusion model cited in the relevant paper is used. For instance, ZipLoRA Shah et al. (2023) employs SDXL, while MixOfShow Gu et al. (2023) utilizes EDLoRA alongside SDv1.5. Similarly, our method uses SDv1.5. Note that while the majority of our concepts are derived from CustomConcept101 dataset, the contribution of our benchmark LoRA collection is the 131 LoRA models and additional 200 text prompts.

---

[5]`https://web.archive.org/web/20231025170948/https://semicolon.dev/midjourney/how-to-make-consistent-characters`

### D.2 EXPERIMENTAL PROMPTS

To evaluate the merging capabilities of the methods, we created 200 text prompts designed to represent various scenarios such as (the corresponding LoRA models are indicated within paranthesis):

- A cat and a dog in the mountain (blackcat, browndog)
- A cat and a dog at the beach (blackcat, browndog)
- A cat and a dog in the street (blackcat, browndog)
- A cat and a dog in the forest (blackcat, browndog)
- A plushie bunny and a flower in the forest (plushie_bunny and flower_1)
- A cat and a flower on the mountain (blackcat, flower_1)
- A cat and a chair in the room (blackcat, furniture_1)
- A cat watching a garden scene intently from behind a window, eager to explore. (blackcat, scene_garden)
- A cat playfully batting at a Pikachu toy on the floor of a child's room. (blackcat, toy_pikachu1)
- A cat cautiously approaching a plushie tortoise left on the patio. (blackcat, plushie_tortoise)
- A cat curiously inspecting a sculpture in the garden, adding to the scenery. (blackcat, scene_sculpture1)

## E COMPARISON WITH LORA-COMPOSER

We compare CLoRA with LoRA-Composer, which operates at test time but requires user-provided bounding boxes, significantly limiting its practicality and ease of use. Additionally, LoRA-Composer is restricted to specific models like ED-LoRA and is incompatible with the wide range of community LoRAs available on platforms like Civit.ai. It also demands substantially more memory, requiring 60GB for generating a composition compared to our method's 25GB for composing two LoRA models. In contrast, CLoRA works seamlessly with any standard LoRA models, including community-sourced ones, without relying on bounding boxes or additional conditions. As shown in Fig. 12, CLoRA consistently produces coherent multi-concept compositions, even in challenging scenarios, ensuring broader compatibility and efficiency. For Fig. 12, the same seed was used for LoRA-Composer with and without bounding boxes to demonstrate the impact of their presence on the results.

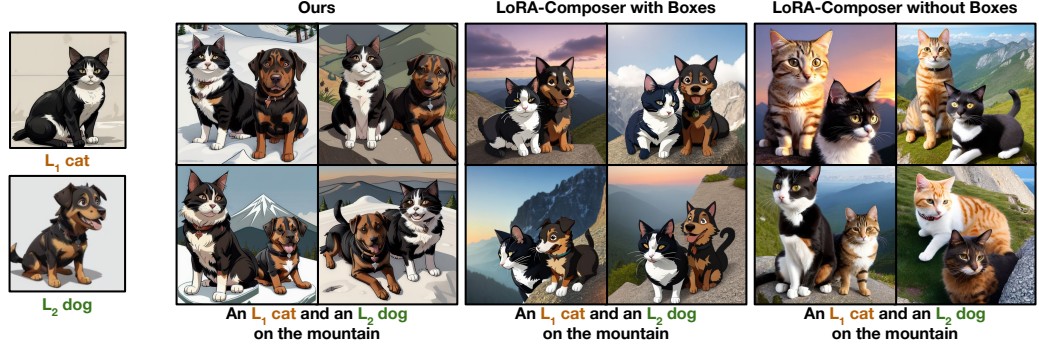

Figure 12: **Qualitative comparison with LoRA-Composer.** CLoRA achieves consistent multi-concept compositions without bounding boxes, unlike LoRA-Composer. Without user-provided bounding boxes, LoRA-Composer method fails to generate the accurate depictions (see rightmost images).

## F ADDITIONAL QUANTITATIVE ANALYSIS

In addition to the results presented in the main paper, we apply further experiments to assess the performance of our method in detail. Specifically, we apply instance segmentation methods to the composed

|  |  | Merge | Composite | ZipLoRA | Mix-of-Show | **Ours** |
|---|---|---|---|---|---|---|
| **CLIP** | Min. | 76.0% ± 8.7% | 76.2% ± 7.2% | 73.4% ± 8.1% | 75.2% ± 9.5% | **83.3% ± 5.5%** |
| | Avg. | 79.5% ± 8.3% | 79.7% ± 6.8% | 77.1% ± 7.6% | 78.7% ± 9.2% | **87.1% ± 4.9%** |
| | Max. | 82.5% ± 8.1% | 82.5% ± 6.7% | 80.6% ± 7.6% | 81.7% ± 9.2% | **89.8% ± 4.8%** |
| **DINO** | Min. | 37.0% ± 15% | 30.3% ± 13% | 36.9% ± 13% | 37.5% ± 17% | **47.2% ± 14%** |
| | Avg. | 43.7% ± 17% | 38.5% ± 13% | 49.6% ± 15% | 48.0% ± 22% | **57.3% ± 14%** |
| | Max. | 50.5% ± 17% | 49.5% ± 14% | 53.3% ± 16% | 55.6% ± 23% | **69.1% ± 14%** |

images to identify and isolate object instances. For this, we use SEEM (Zou et al., 2024) to segment the objects within the images. After segmentation, we calculate the similarity metrics separately for each object instance, allowing for a more granular comparison of the methods. We perform these evaluations on a set of 700 images per method, as shown in the table. The results demonstrate that our method significantly outperforms others across multiple metrics. In particular, we calculate DINO scores, which further highlight the effectiveness of our approach compared to competing methods. Moreover, we also compute CLIP scores as additional evidence of our method's superior performance.

## G  ADDITIONAL QUALITATIVE RESULTS

**Comparison with OMG.**   We perform a qualitative comparison between our method, CLoRA, and OMG (Kong et al., 2024). OMG relies on off-the-shelf segmentation methods to isolate subjects before generating images. As seen in Fig. 13, while this enables well-defined subject boundaries, the performance of OMG is heavily dependent on the accuracy of the segmentation model. Errors in segmentation can result in incomplete or incorrect generation, particularly in complex scenes involving multiple interacting subject. For instance, if the segmentation model fails to detect a flower, this may prevent the correct placement of the LoRA in the composition (see Fig. 13 bottom-left). Moreover, since OMG depends on the base image generated by the Stable Diffusion model, it also encounters the attention overlap and attribute binding issues identified by Chefer et al. (2023). For instance, if the Stable Diffusion model does not generate the required objects in the base image from the text prompt 'A man and a bunny in the room', then OMG cannot produce the desired composition. This issue is apparent in Fig. 13, where the rightmost image shows that the base model generated only a bunny, omitting the man. In contrast, CLoRA bypasses the need for explicit segmentation by directly updating attention maps and fusing latent representations. This ensures that each concept, represented by different LoRA models, is accurately captured and preserved during generation. The comparison in Fig. 13 demonstrates that CLoRA produces more coherent compositions, maintaining the integrity of each concept even in challenging multi-concept scenarios.

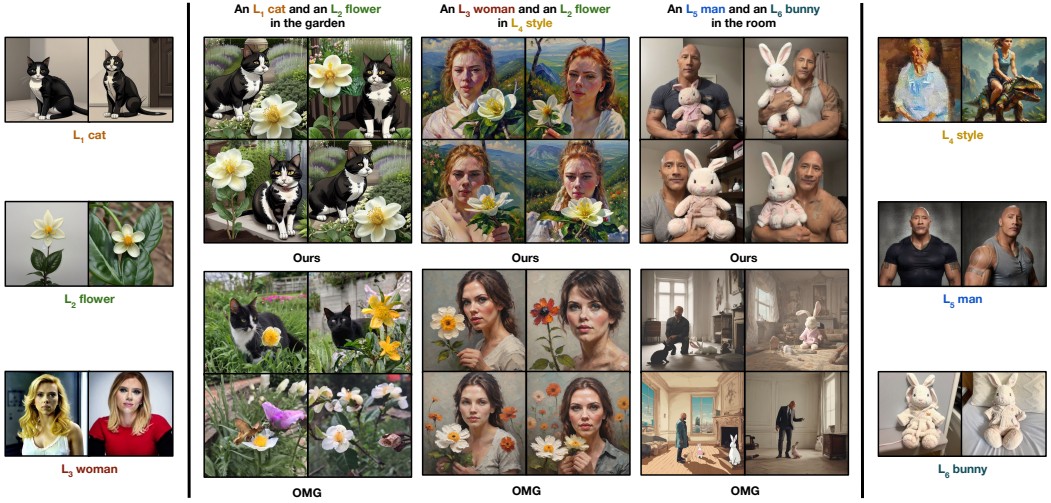

Figure 13: **Qualitative comparison with OMG.** Our method (top row) consistently produces more coherent and accurate compositions compared to OMG (bottom row). By leveraging attention map updates and latent fusion, CLoRA effectively handles multi-concept generation without relying on segmentation, leading to higher quality results, particularly in complex scenes.

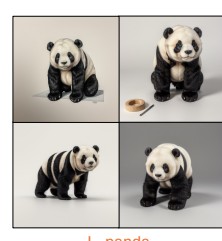 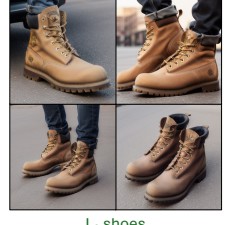 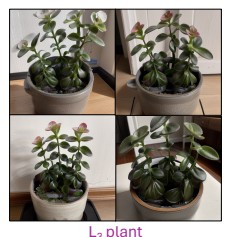

L₁ panda      L₂ shoes      L₃ plant

An L₁ panda, an L₂ shoes and an L₃ plant in the room

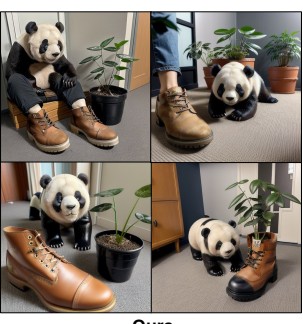 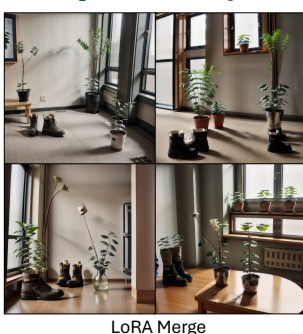 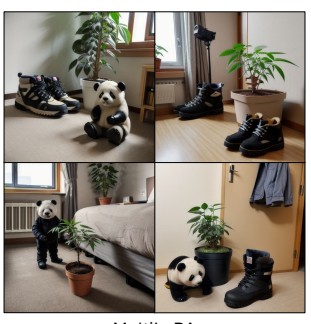

**Ours**      LoRA Merge      MultiLoRA

Figure 14: **Qualitative comparison of CLoRA** with other LoRA methods using 3 LoRAs to generate a single image. Our approach consistently produces images that more accurately reflect the input text prompts, LoRA subjects, and LoRA styles.

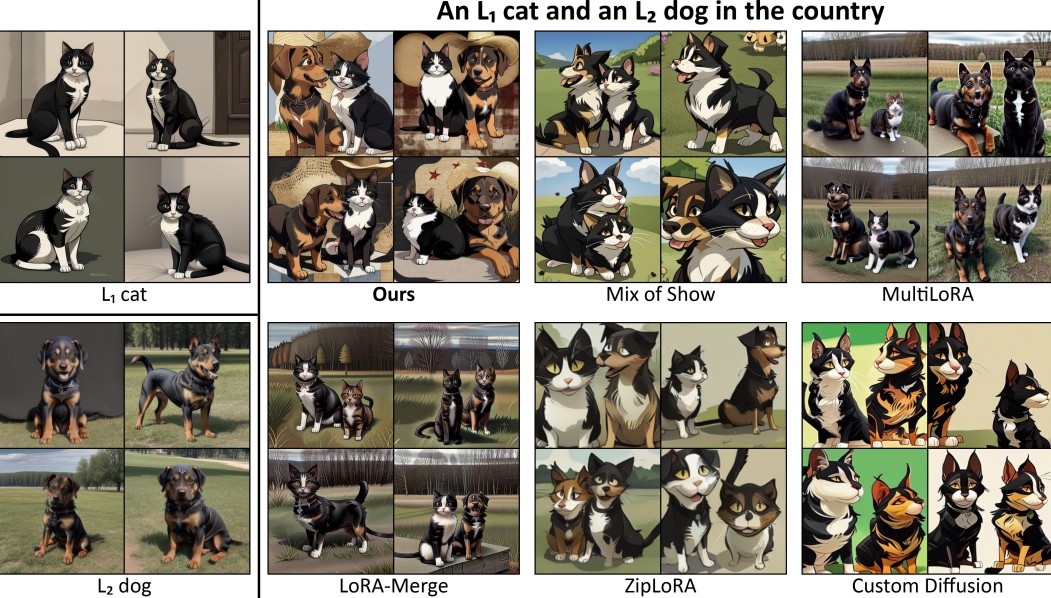

**An L₁ cat and an L₂ dog in the country**

L₁ cat      **Ours**      Mix of Show      MultiLoRA

L₂ dog      LoRA-Merge      ZipLoRA      Custom Diffusion

Figure 15: **Qualitative comparison of CLoRA** with other LoRA methods. Our approach consistently produces images that more accurately reflect the input text prompts, LoRA subjects, and LoRA styles.

**Extensive Qualitative Results.** The rest of the Supplementary Materials will provide additional qualitative comparisons which contain the following competitors: Mix of Show Gu et al. (2023), MultiLoRA Zhong et al. (2024), LoRA-Merge Ryu (2023), ZipLoRA Shah et al. (2023), and Custom Diffusion Kumari et al. (2023) on various LoRAs and prompts. Figure 14 compare LoRA-Merge and MultiLoRA using three combined LoRAs, while later figures expand the comparison to include all methods across two separate LoRAs.

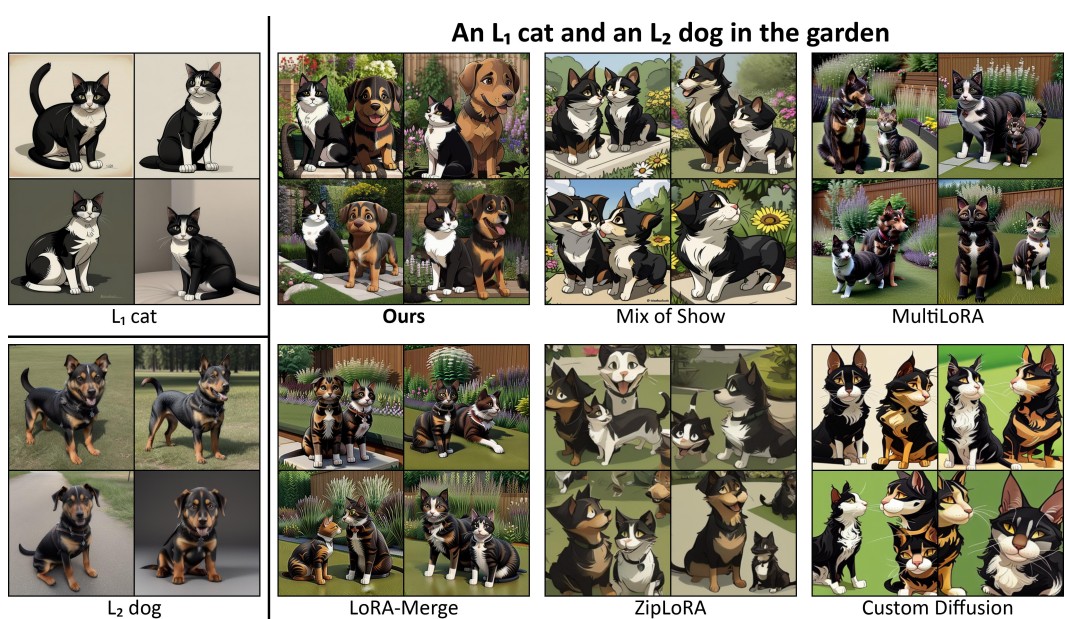

Figure 16: **Qualitative comparison of CLoRA** with other LoRA methods. Our approach consistently produces images that more accurately reflect the input text prompts, LoRA subjects, and LoRA styles.

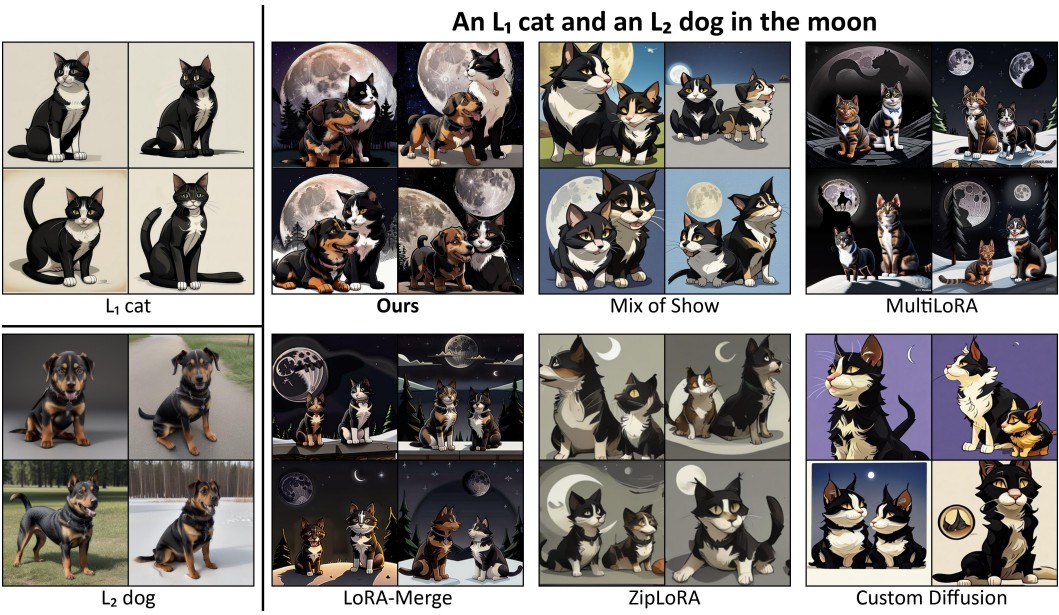

Figure 17: **Qualitative comparison of CLoRA** with other LoRA methods. Our approach consistently produces images that more accurately reflect the input text prompts, LoRA subjects, and LoRA styles.

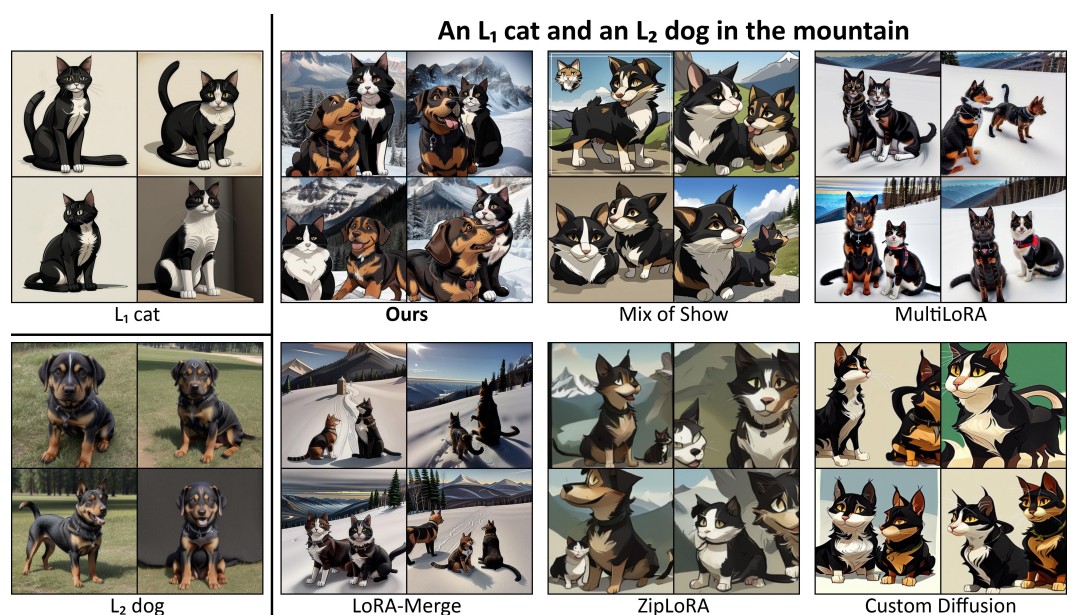

Figure 18: **Qualitative comparison of CLoRA** with other LoRA methods. Our approach consistently produces images that more accurately reflect the input text prompts, LoRA subjects, and LoRA styles.

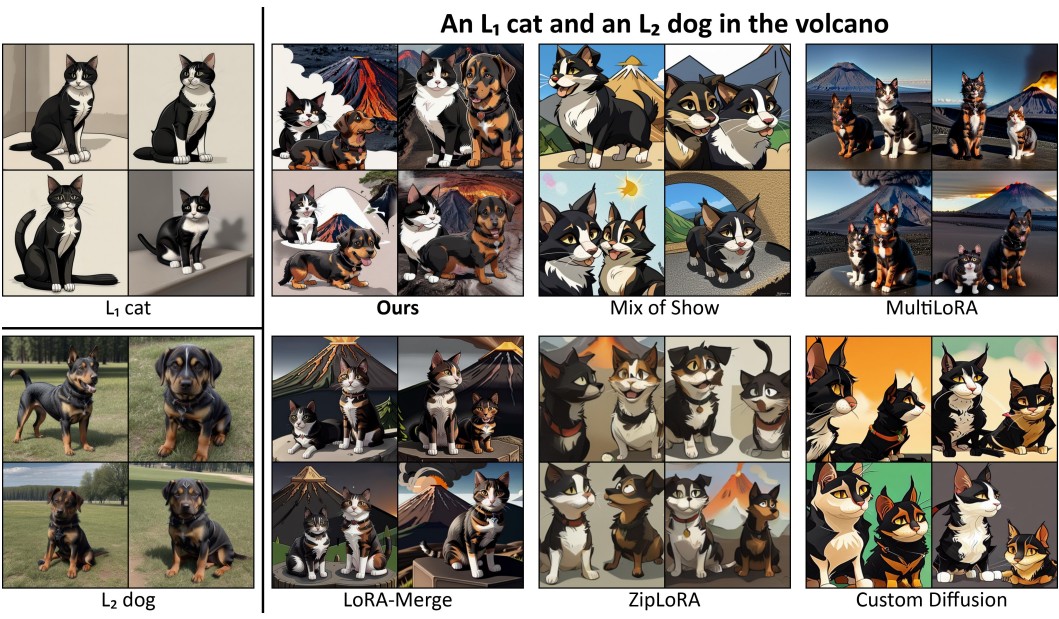

Figure 19: **Qualitative comparison of CLoRA** with other LoRA methods. Our approach consistently produces images that more accurately reflect the input text prompts, LoRA subjects, and LoRA styles.

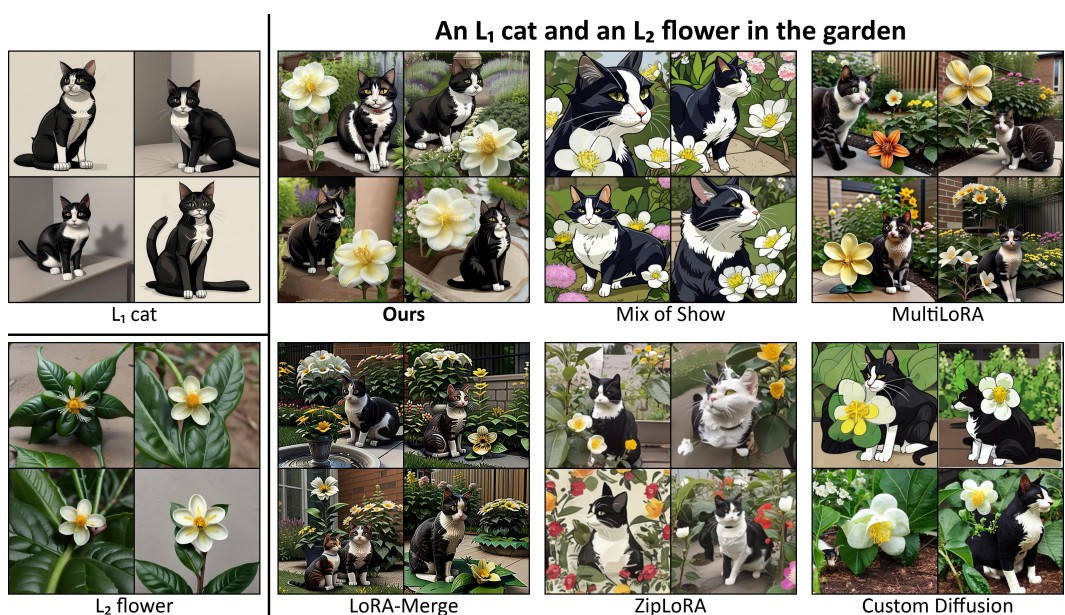

Figure 20: **Qualitative comparison of CLoRA** with other LoRA methods. Our approach consistently produces images that more accurately reflect the input text prompts, LoRA subjects, and LoRA styles.

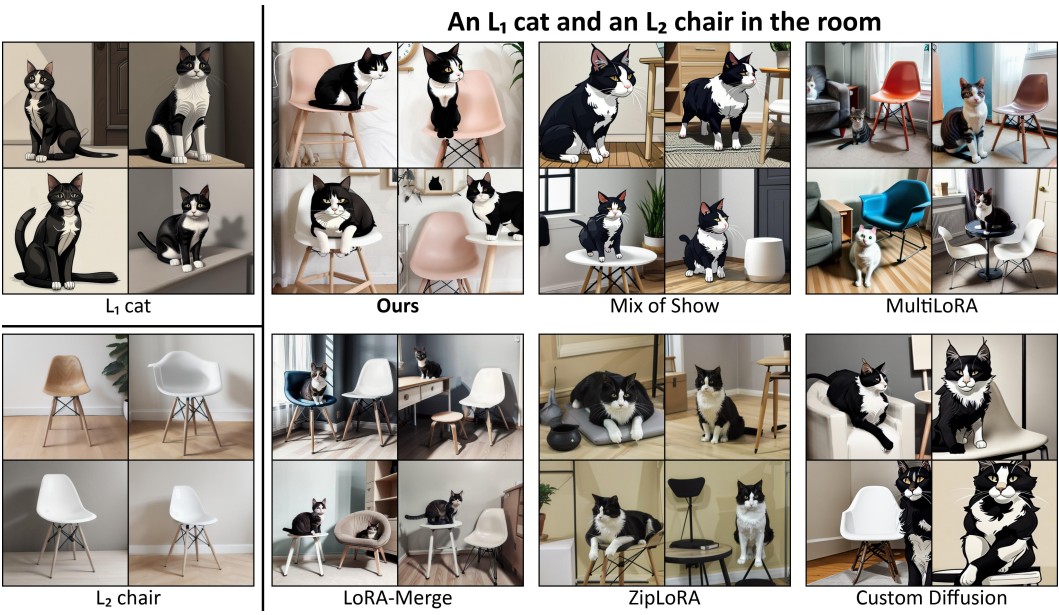

Figure 21: **Qualitative comparison of CLoRA** with other LoRA methods. Our approach consistently produces images that more accurately reflect the input text prompts, LoRA subjects, and LoRA styles.

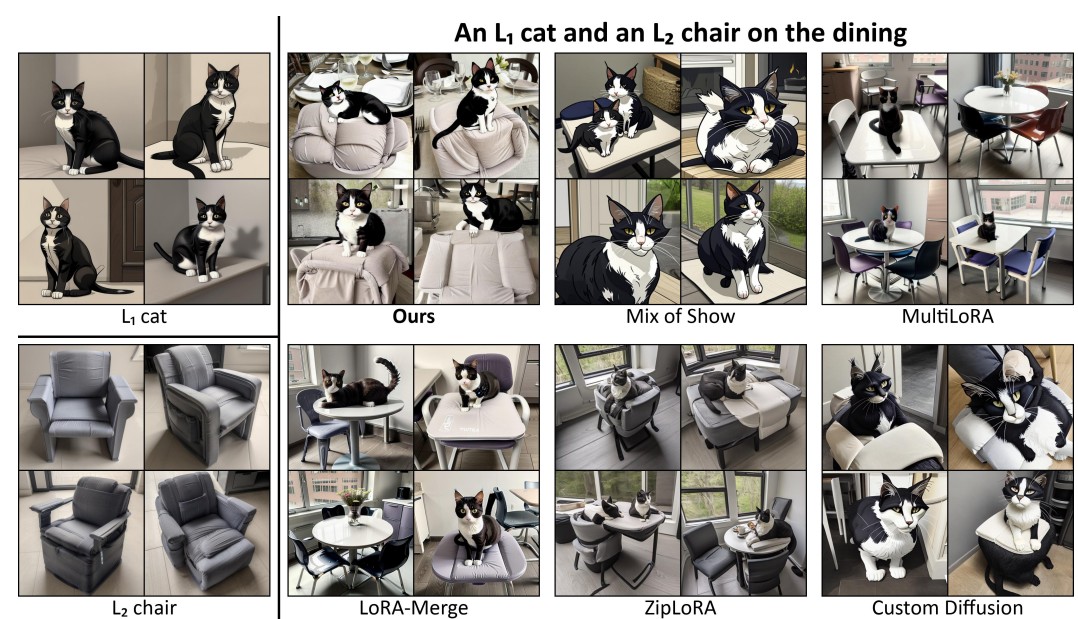

Figure 22: **Qualitative comparison of CLoRA** with other LoRA methods. Our approach consistently produces images that more accurately reflect the input text prompts, LoRA subjects, and LoRA styles.

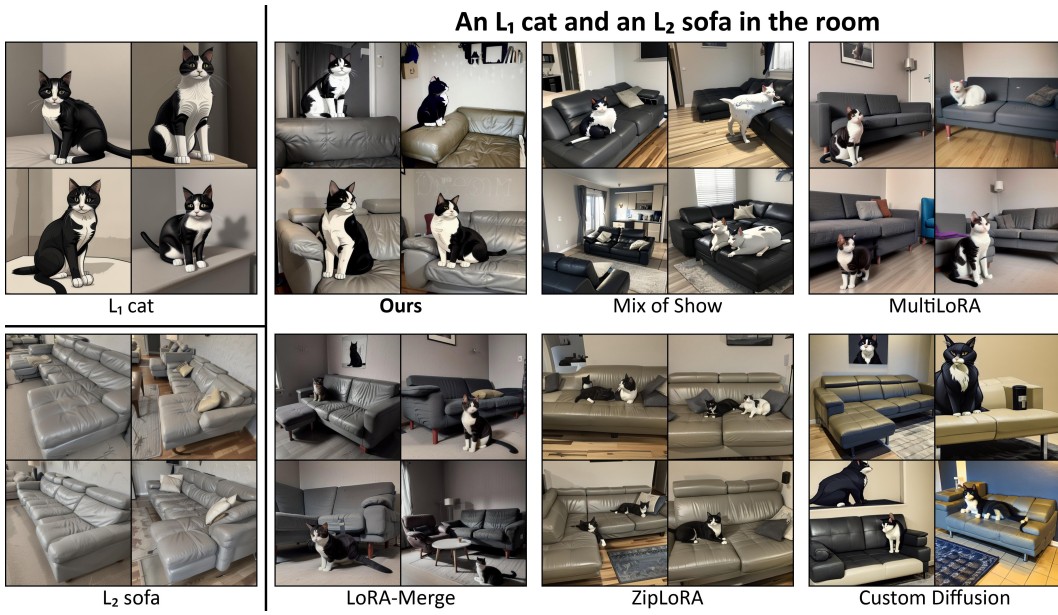

Figure 23: **Qualitative comparison of CLoRA** with other LoRA methods. Our approach consistently produces images that more accurately reflect the input text prompts, LoRA subjects, and LoRA styles.

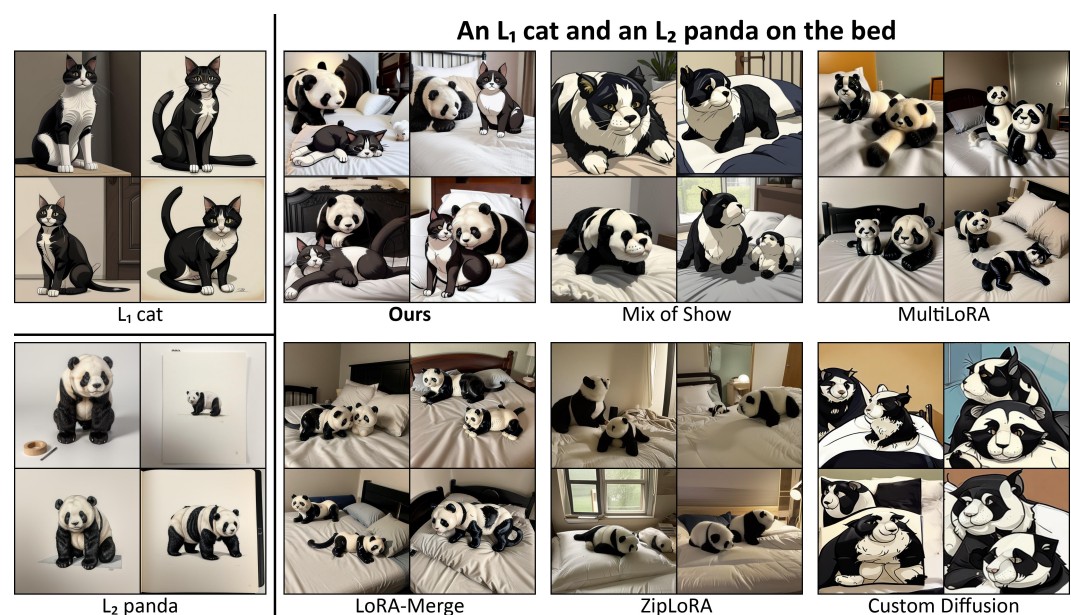

Figure 24: **Qualitative comparison of CLoRA** with other LoRA methods. Our approach consistently produces images that more accurately reflect the input text prompts, LoRA subjects, and LoRA styles.

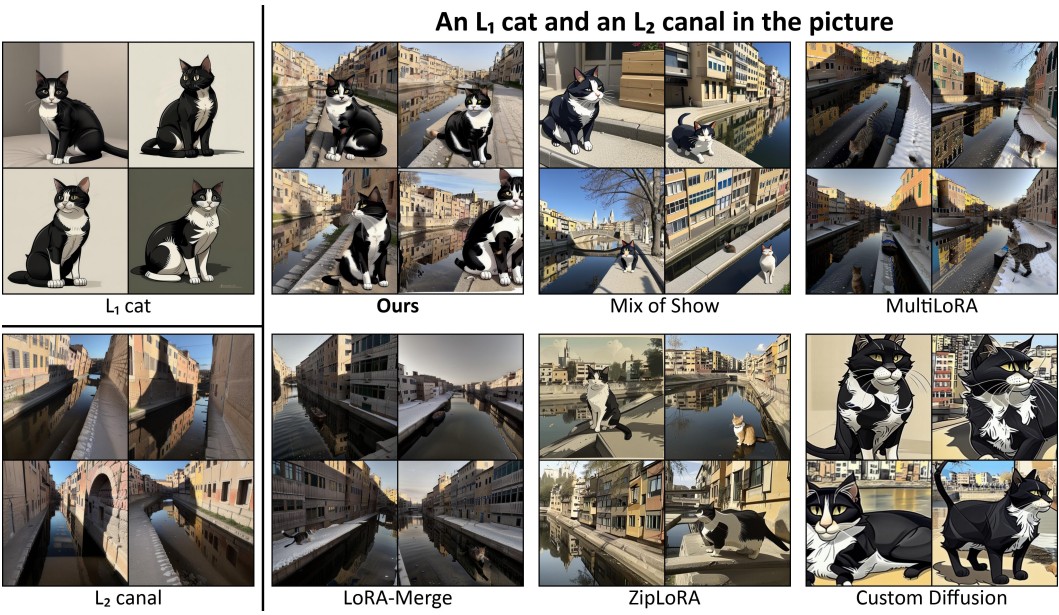

Figure 25: **Qualitative comparison of CLoRA** with other LoRA methods. Our approach consistently produces images that more accurately reflect the input text prompts, LoRA subjects, and LoRA styles.

**An L₁ cat and an L₂ sculpture in the garden**

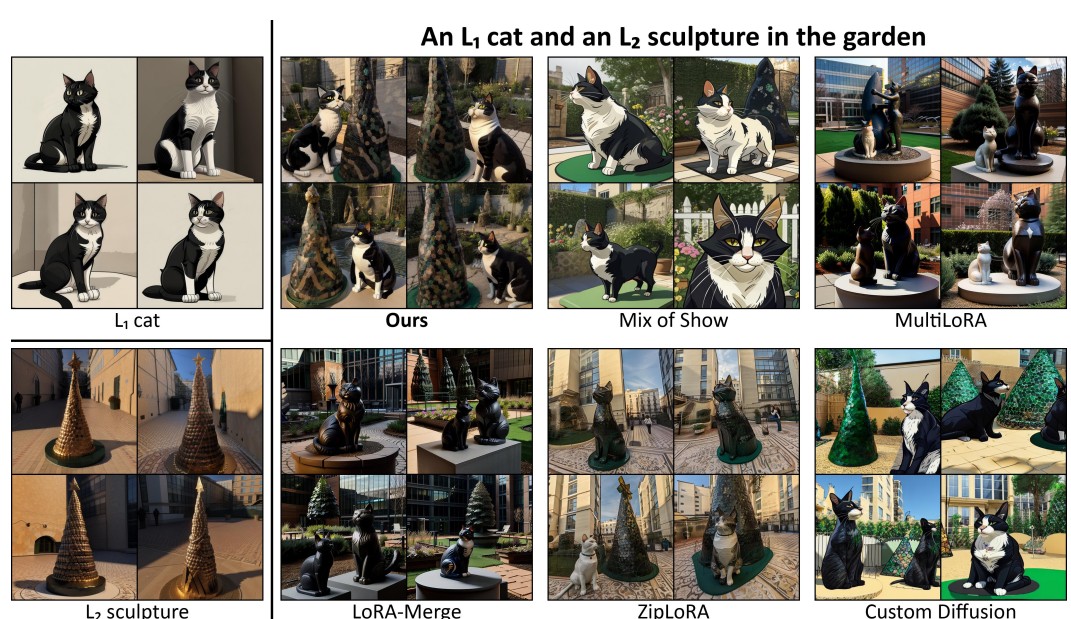

Figure 26: **Qualitative comparison of CLoRA** with other LoRA methods. Our approach consistently produces images that more accurately reflect the input text prompts, LoRA subjects, and LoRA styles.

**An L₁ cat and an L₂ bike in the garage**

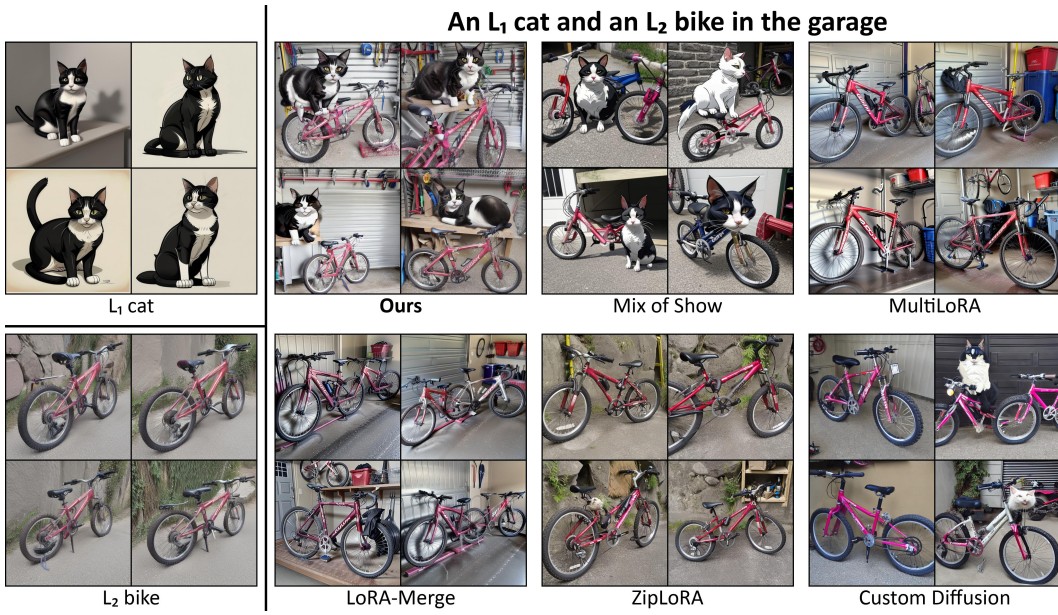

Figure 27: **Qualitative comparison of CLoRA** with other LoRA methods. Our approach consistently produces images that more accurately reflect the input text prompts, LoRA subjects, and LoRA styles.

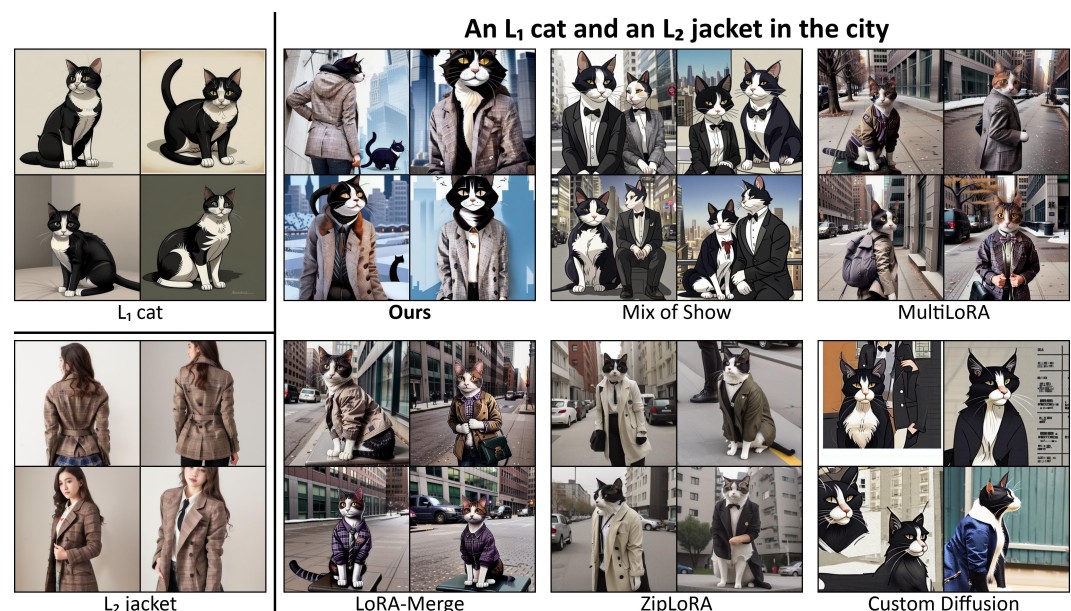

Figure 28: **Qualitative comparison of CLoRA** with other LoRA methods. Our approach consistently produces images that more accurately reflect the input text prompts, LoRA subjects, and LoRA styles.

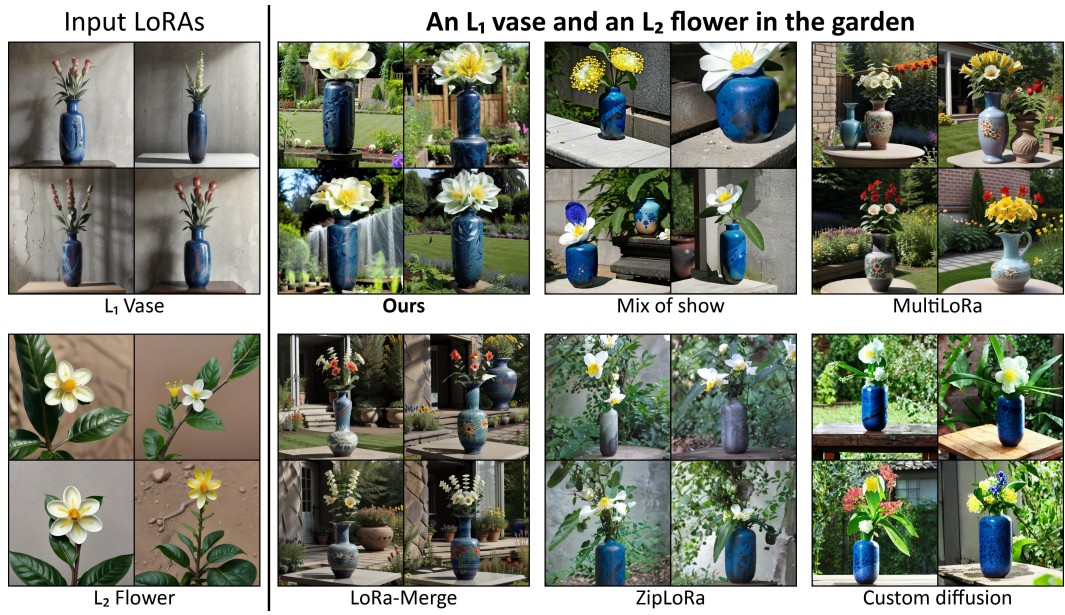

Figure 29: **Qualitative comparison of CLoRA** with other LoRA methods. Our approach consistently produces images that more accurately reflect the input text prompts, LoRA subjects, and LoRA styles.

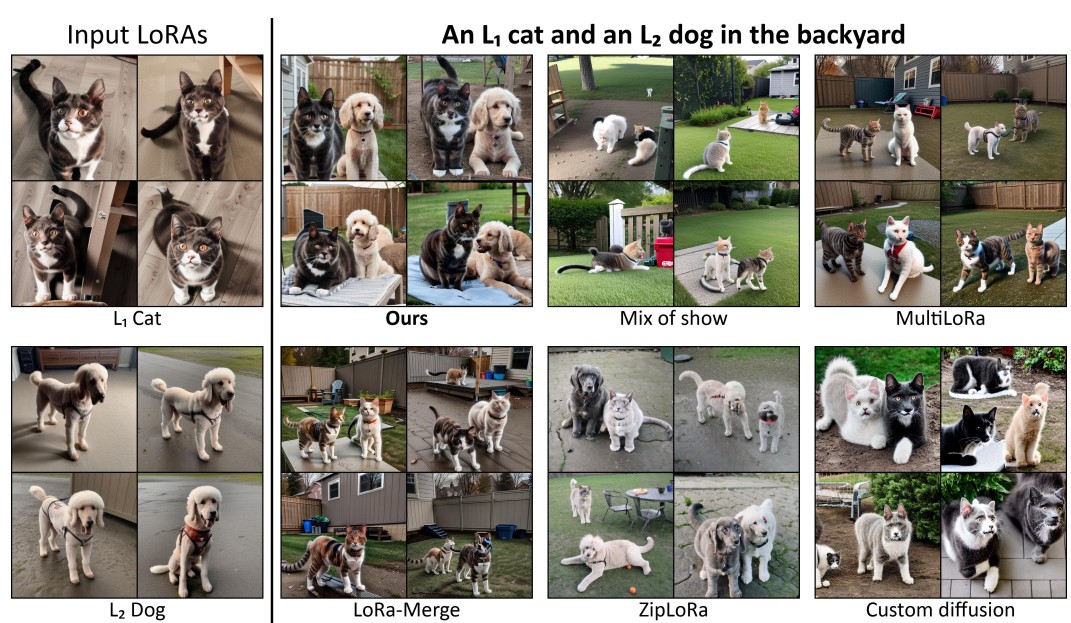

Figure 30: **Qualitative comparison of CLoRA** with other LoRA methods. Our approach consistently produces images that more accurately reflect the input text prompts, LoRA subjects, and LoRA styles.

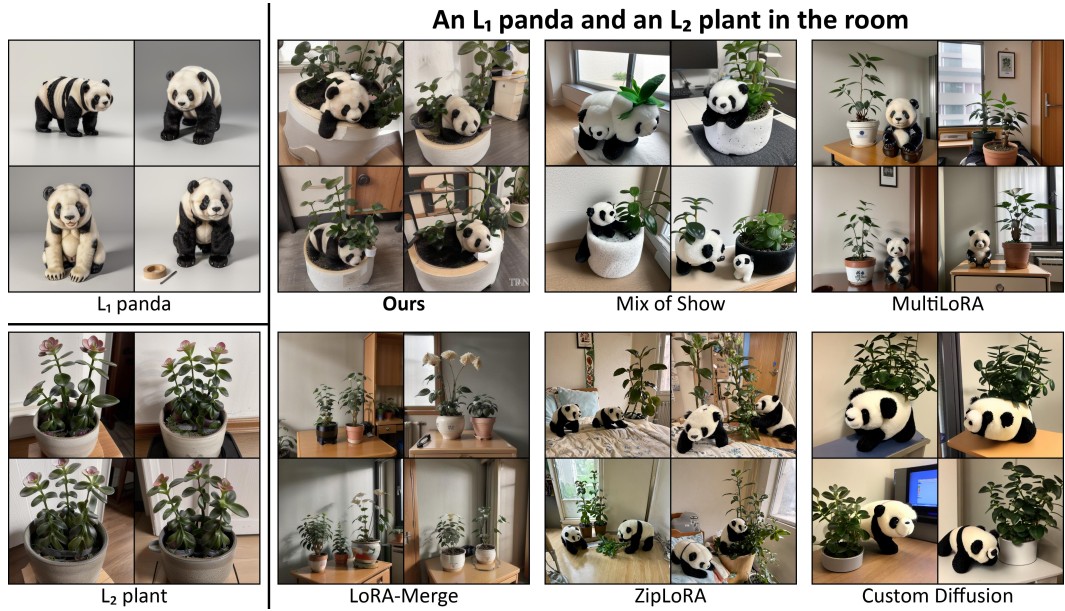

Figure 31: **Qualitative comparison of CLoRA** with other LoRA methods. Our approach consistently produces images that more accurately reflect the input text prompts, LoRA subjects, and LoRA styles.

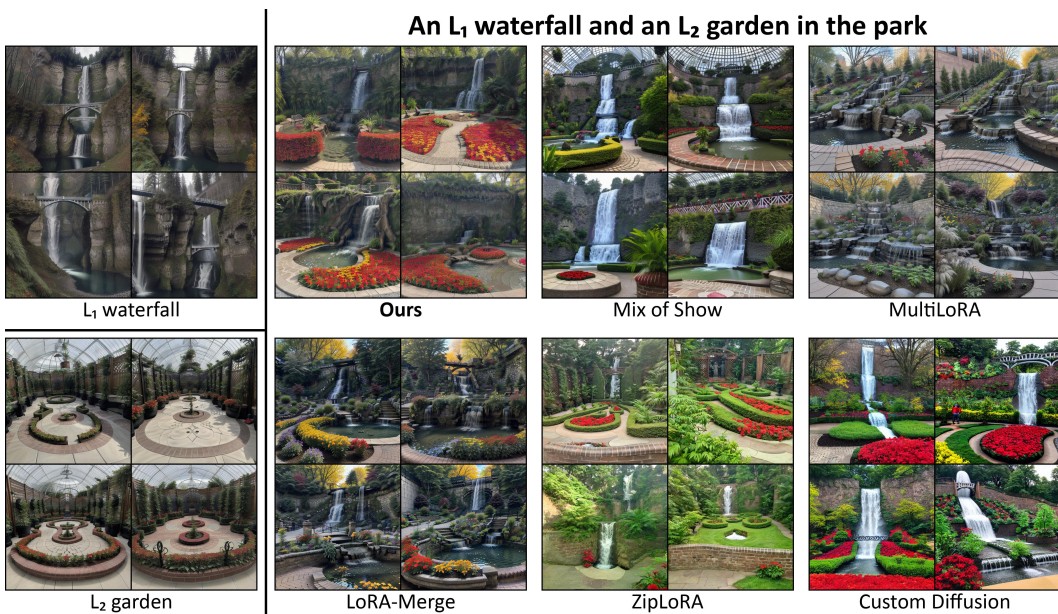

Figure 32: **Qualitative comparison of** `CLoRA` with other LoRA methods. Our approach consistently produces images that more accurately reflect the input text prompts, LoRA subjects, and LoRA styles.

