# OpenReview forum: "CLoRA: A Contrastive Approach to Compose Multiple LoRA Models"
_ICLR.cc/2025/Conference — Submitted to ICLR 2025_

### Official Review · Reviewer_EEUA · 2024-10-19

**Soundness:** 3
**Presentation:** 3
**Contribution:** 3
**Rating:** 6
**Confidence:** 4

**Summary:**

The paper introduces CLoRA, an approach to enhance the use of Low-Rank Adaptation (LoRA) models for multi-concept image generation. CloRA aims to address the limitations of existing methods, which struggle with overlapping attention mechanisms in multiple LoRA models. CloRA updates attention maps at test-time, using these maps to create semantic masks for better fusion of latent representations. The authors claim that CloRA significantly outperforms existing methods in generating composite images that accurately reflect the characteristics of each LoRA. They also provide a benchmark dataset to facilitate further research.

**Strengths:**

1. The paper addresses a relevant and challenging problem in the field of personalized image generation, making it a meaningful contribution to the community.
2. The authors provide a benchmark dataset, which is valuable for systematic study and future research.
3. The paper is well-structured and clearly written, making it easy to follow and understand the proposed approach and its benefits.

**Weaknesses:**

1. The author should report the time and computational consumption of their test-time adaptation strategy.  The training-free attribute is less attractive if it takes a similar amount of time or memory usage to methods that require fine-tuning.
2. The technical novelty of this work requires further justification. Adjusting attention weights is a common strategy to alleviate the problems of concept ignorance or incorrect combination, and has been studied in many works [1,2,3,4].
3. Missing Citation and Comparison. The authors are suggested to make comparisons with Lora-Composer [4] which also targets multi-LoRA composition and adjusts attention weights through test-time optimization.
4. The authors are suggested to make a comprehensive evaluation of the scalability of the method. For example, what is the maximum number of LoRA models the proposed method can handle? Are there any trends in performance as the number of LoRA models increases?

[1] Attend-and-Excite: Attention-Based Semantic Guidance for Text-to-Image Diffusion Models

[2] Grounded Text-to-Image Synthesis with Attention Refocusing

[3] BoxDiff: Text-to-Image Synthesis with Training-Free Box-Constrained Diffusion

[4] LoRA-Composer: Leveraging Low-Rank Adaptation for Multi-Concept Customization in Training-Free Diffusion Models

**Questions:**

Please refer to weaknesses.

---

> ### Author Response · Authors · 2024-11-23
>
> Thank you for the supportive feedback. We respond to your comments below in two separate answers due to space constraints. Please feel free to let us know if you have any further questions.
>
> **Efficiency and Scalability Analysis:**  Thanks for pointing this out. We run a comparison between competing methods and ours in terms of time consumption when generating a composition. *Our test-time optimization takes 24 seconds* in an Nvidia H100 GPU  for image composition using 2 LoRA models, while *other models that require finetuning or training consume an additional 5-15 minutes* before they can generate the composition. Moreover, while methods such as LoRA Merge or Multi-LoRA seem more efficient in terms of runtime, our results (Figures 3–5, Table 1) demonstrate much better performance in both qualitative and quantitative metrics for handling LoRA compositions efficiently. *We revised our manuscript with an in-depth discussion. More details can be found in Appendix A.1, and Table 2  in our new manuscript.*
>
> | Method                | CivitAI Compatibility | VRAM (Finetuning/Inference) | Runtime (Finetuning/Inference) |
> |-----------------------|-----------------------|-----------------------------|---------------------------------|
> | Custom Diffusion      | no                   | 28GB + 8 GB                 | 4.2 min + 3.5s                 |
> | LoRA Merge            | yes                  | 7 GB                        | 3.2 s                          |
> | Multi-LoRA - composite| yes                  | 7 GB                        | 3.4 s                          |
> | Multi-LoRA - switch   | yes                  | 7 GB                        | 4.8 s                          |
> | Mix-of-Show           | no                   | 10GB + 10 GB                | 10min + 3.3s                   |
> | ZipLoRA               | yes                  | 39GB + 17 GB                | 8min + 4.2s                    |
> | OMG                   | yes                  | 30 GB                       | 62 s                           |
> | LORA Composer         | no                   | 51 GB                       | 35 s                           |
> | Ours                  | yes                  | 25 GB                       | 24 s                           |
>
>
>
> **Technical Novelty Requires Justification:** Thank you for your comment. As highlighted in our paper (Lines 097-099), prior research has identified attention overlap and attribute binding issues as challenges in image generation tasks. *Our novel observation, presented for the first time in this paper, is that these issues also arise in LoRA composition tasks.* Specifically, when multiple LoRA models are composed, their attentions can overlap, leading to failures in generating the intended compositions. While this observation is novel, the *core technical innovation of our work lies in the proposed contrastive objective function.* This objective effectively separates the attention of multiple LoRA models during test-time, without requiring any additional training, user-provided masking, or conditions. To the best of our knowledge, both the observation regarding attention overlap in LoRA composition and the design of our method are novel contributions. Moreover, both our visual and quantitative results (Figures 3–5, Table 1) demonstrate superior performance for handling LoRA compositions compared to competing methods, showing the efficiency of the proposed method.
>
> **Missing citation and comparison:** Thank you for bringing these related works to our attention. *We included a comparison in our revised manuscript* (please see Section 2). Furthermore, we would like to clarify the key differences between our method and LoRA-Composer. While LoRA-Composer also operates at test time, *it requires user-provided bounding boxes as additional conditions*, which significantly limits its practicality and ease of use. Additionally, *their method is restricted to specific LoRA models, such as ED-LoRA*, and is not compatible with the wide range of community LoRAs available on platforms like Civit.ai. Moreover, their method has substantially higher memory requirements, *needing 60GB of memory for generating a composition*, as noted in their [GitHub repository](https://github.com/Young98CN/LoRA_Composer?tab=readme-ov-file#wrench-dependencies-and-installation), while our method only requires 25GB for composing two LoRA models. To provide a comprehensive evaluation, we included a comparison with LoRA-Composer in Appendix E and Figure 12. As shown in the results, while LoRA-Composer achieves comparable performance when user-provided bounding boxes are available, *it fails to generate coherent compositions without them*. In contrast, our method does not require any additional conditions, such as bounding boxes, and works seamlessly with any standard LoRA models, including those available on community platforms like Civit.ai.
>
> **Please see the next answer for other comments.**

---

> > ### Author Response · Authors · 2024-11-23
> >
> > **Evaluation of the scalability of the method:** Thank you for this suggestion. We conducted additional evaluations on the model’s scalability with an increasing number of LoRA compositions. *Please refer to Appendix A.2 and Figure 7 of the revised manuscript, where we demonstrate how memory usage and runtime increase with the number of LoRA compositions.* Our method's runtime ranges from 24 seconds (for 2 LoRAs) to 95 seconds (for 8 LoRAs), while memory consumption ranges from 25 GB to 80 GB. We would like to highlight that supporting compositions with a large number of LoRA models is a *challenging task for all competitor methods as well.* This is primarily due to the memory consumption of individual LoRA models and the difficulty in effectively separating multiple LoRA concepts without blending them. Upon further examination of competing methods, we observed that ZipLoRA showcased compositions up to 2 LoRA models, MultiLoRA up to 3 LoRAs, and Mix of Show up to 5 LoRA models. It is important to note, however, that Mix of Show requires additional inputs from the user, such as human pose or canny edges, whereas our method operates without these conditions.

---

> > > ### Comment · Reviewer_EEUA · 2024-11-27
> > >
> > > Thanks for the rebuttal. It addresses most of my concerns. CLoRA's doesn't impress me with its innovativeness, but I believe it can make a positive contribution to the image generation community. For this reason, I am willing to raise my score to 6.

---

> > > > ### Author Response · Authors · 2024-12-01
> > > >
> > > > Dear Reviewer EEUA,
> > > >
> > > > Thank you for acknowledging the positive contribution of our paper to the image generation community and for your decision to increase your score. We truly appreciate your time and constructive feedback, which have greatly improved the quality of our revised manuscript.
> > > >
> > > > Best regards,
> > > >
> > > > Authors of Paper 3307

---

### Official Review · Reviewer_KmNX · 2024-10-28

**Soundness:** 2
**Presentation:** 3
**Contribution:** 2
**Rating:** 5
**Confidence:** 5

**Summary:**

This paper addresses the challenges of using multiple Low-Rank Adaptation models for personalized image generation, where combining multiple LoRAs often leads to issues like attention overlap and attribute binding. To resolve this, the paper proposes CLoRA, a training-free method that updates attention maps at test-time, ensuring each LoRA model focuses on its respective concept. CLoRA improves image generation by refining attention maps and using semantic masks to blend representations, outperforming existing methods. It works without specialized LoRA variants or training and also includes a benchmark dataset for evaluation.

**Strengths:**

1.	The paper proposes a training-free method to merge LoRAs.
2.	It introduces a diverse benchmark dataset comprising various LoRA models
3.	The work is complete, providing a new contrastive approach CLoRA and a benchmark dataset. CLoRA achieves better performance compared to previous methods.
4.	This is the first comprehensive work to observe and address attention overlap and attribute binding within LoRA-enhanced image generation models, which offers a new perspective of attention map for this task.

**Weaknesses:**

1. Writing weakness:
   1. The style is not uniform(line 263-264 $L_{1}$-applied, $L_{2}$-applied and line 323 $L_1$). It is same as $L_1$ in red color?
   2. The meaning of "N" in equation 2 is unclear.
2. Poor visualization quality:
   1. In Figure 3, the visual similarity between the animals and the reference images is lacking, and it seems to exhibit an animated style without applying the style tokens indicated in the prompts.
   2. In Figure 4, the depiction of the book loses essential characteristics, and the cup does not closely resemble the reference images.
3. Insufficient evaluation metrics: The paper only considers image similarity, neglecting text similarity, which is commonly evaluated in baseline papers.
4.	Lack of analysis of the model efficiency. Efficiency is naturally crucial for test-time approach, but the analysis is absent.

**Questions:**

1. Does the approach work in cases where two similar concept categories are generated (e.g., “catA” and “catB”)? I am concerned that the contrastive loss might fail in such scenarios.
2. In the results depicted in Figure 4 for the Mix-of-Show, are conditions such as human pose or canny edge used as inputs? If so, could you please describe how these input conditions are generated?
3.	Are there more detailed analysis of the model efficiency to prove its test-time usability?
4.	Are there some small errors of the subscripts of $Z$s in Figure 2?

---

> ### Author Response · Authors · 2024-11-23
>
> Thank you for the supportive feedback. We respond to your comments below in **two separate answers** due to space limitations. Please feel free to let us know if you have any further questions.
>
> **Lack of Uniform Notation and Typos:** Thanks for bringing these to our attention. We used the same color convention for contrastive token pairs to facilitate easier reading while explaining the intuition behind our method. *We have revised our manuscript* to clarify this notation in Lines 249-250 and have also corrected the mentioned typos. Please refer to the updated version we have uploaded, where the changes are highlighted in green.
>
> **Poor Visualization Quality/Animated Style in Fig 3 and Lack of Resemblance in Fig 4:**  *Regarding the Animated Style in Fig 3:* We apologize for the error in Figure 3, where the original LoRA model for the dog example was depicted incorrectly. The correct version of this LoRA model, which indeed has an animation-style appearance, is shown in our teaser image, Figure 1. Therefore, our depictions in our LoRA compositions are accurate. *We updated Fig 3  to reflect the correct image in our updated manuscript.* Moreover, as can be seen from our results in both Fig 3 and Fig 4, when LoRA models depict realistic images such as a realistic shoe, plant, or bicycle, *our compositions reflect these objects realistically.*
>
> *Regarding the example that depicts the cup and book objects together in Figure 4:* we acknowledge that some essential characteristics of the book might not be closely depicted to the original LoRA model. This challenge is likely due to the quality of the original LoRA model itself. It's important to note that none of the five competing methods performed this composition correctly; they either generated completely different cup or book objects or blended the book cover with the cup object. Compared to the competitors, our method successfully generated both objects without blending and with a closer resemblance than the other methods. *We added a clarification about this example to the revised manuscript, please see lines 429-431.*
>
> **Additional Evaluation Metrics:** Thank you for this suggestion. We computed CLIP-I and CLIP-T scores, as can be seen from the following table. Note that our method performed the highest scores for both CLIP-T and CLIP-I scores. These results also support the DINO metrics presented in the main paper. *We revised our manuscript to include these metrics, along with a discussion.* Please see our revised manuscript, Table 1.
>
> |                |               | Merge [Ryu et al., 2023] | Composite | Switch [Zhong et al., 2024] | ZipLoRA [Shah et al., 2023] | Mix-of-Show [Gu et al., 2023] | **Ours**         |
> |----------------|---------------|--------------------------|-----------|-----------------------------|-----------------------------|-------------------------------|------------------|
> | **CLIP-I**     | **Min.**      | 0.641 ± 0.029           | 0.614 ± 0.035 | 0.619 ± 0.039           | 0.659 ± 0.022           | 0.664 ± 0.023               | **0.683 ± 0.017** |
> |                | **Avg.**      | 0.683 ± 0.029           | 0.654 ± 0.035 | 0.659 ± 0.036           | 0.707 ± 0.021           | 0.712 ± 0.022               | **0.725 ± 0.017** |
> |                | **Max.**      | 0.714 ± 0.028           | 0.690 ± 0.033 | 0.695 ± 0.036           | 0.740 ± 0.021           | 0.744 ± 0.023               | **0.756 ± 0.017** |
> | **CLIP-T**     |               | 0.814 ± 0.054           | 0.833 ± 0.091 | 0.822 ± 0.089           | 0.767 ± 0.081           | 0.760 ± 0.074               | **0.862 ± 0.052** |
>
>
> **Please see our next answer for other comments.**

---

> ### Author Response · Authors · 2024-11-23
>
> **Efficiency Analysis:** Thanks for pointing this out. We run a comparison between competing methods and ours in terms of time consumption when generating a composition. *Our test-time optimization takes 24 seconds* for image composition using 2 LoRA models, while other models that require finetuning or training consume an additional 5-15 minutes before they can generate the composition. Moreover, while methods such as LoRA Merge or MultiLoRA seem more efficient in terms of runtime, our results (Figures 3–5, Table 1) demonstrate much better performance in both qualitative and quantitative metrics for handling LoRA compositions efficiently. *More details can be found in Appendix A, Table 2, and Figure 7 in our revised manuscript.*
>
> | Method                | CivitAI Compatibility | VRAM (Finetuning/Inference) | Runtime (Finetuning/Inference) |
> |-----------------------|-----------------------|-----------------------------|---------------------------------|
> | Custom Diffusion      | no                   | 28GB + 8 GB                 | 4.2 min + 3.5s                 |
> | LoRA Merge            | yes                  | 7 GB                        | 3.2 s                          |
> | Multi-LoRA - composite| yes                  | 7 GB                        | 3.4 s                          |
> | Multi-LoRA - switch   | yes                  | 7 GB                        | 4.8 s                          |
> | Mix-of-Show           | no                   | 10GB + 10 GB                | 10min + 3.3s                   |
> | ZipLoRA               | yes                  | 39GB + 17 GB                | 8min + 4.2s                    |
> | OMG                   | yes                  | 30 GB                       | 62 s                           |
> | LORA Composer         | no                   | 51 GB                       | 35 s                           |
> | Ours                  | yes                  | 25 GB                       | 24 s                           |
>
>
> **Does the approach work in cases where two similar concept categories are generated (e.g., “catA” and “catB”)?:** Thank you for your thoughtful question. While contrastive loss might seem contradictory for similar concepts such as two cat LoRAs, our method is designed to handle such cases by applying contrastive loss to “attention maps” corresponding to distinct tokens, even when those tokens represent similar concepts.  *We included examples of our model depicting two cat LoRAs in the same composition, please see our revised manuscript,* Figure 10. For example, in the case of the prompt "A L1 cat and a L2 cat," where the first cat is depicted by L1 LoRA, and the second cat is depicted by L2 LoRA, the model generates two separate attention maps, one for the “cat” with L1 features and one for the “cat” with L2 features. The contrastive loss is then applied to ensure that these attention maps remain distinct and non-overlapping, effectively separating the two cats in the final image.
> Therefore, *our objective function is robust enough to handle similar concept categories* by focusing on the token-level distinction in the attention maps, ensuring that each concept is properly represented without interference, even in the case of similar categories like "catA" and "catB."
>
> **Conditions for Mix-of-show experiments:** To ensure a fair comparison between methods, we did not use additional control inputs like pose or canny edges for both qualitative and quantitative comparisons. It's important to note that in quantitative comparisons, using such controls would be very prohibitive, as *we would need to generate specific control conditions for each prompt* and LoRA combination in our experiments.

---

> ### Author Response · Authors · 2024-12-01
>
> Dear Reviewer KmNX,
>
> We would like to kindly remind you that tomorrow is the deadline for reviewers to submit their comments. Should you have any further questions or need additional clarification, please do not hesitate to reach out.
>
> We addressed all your concerns (see our rebuttal response above) and revised our manuscript accordingly. **If you find our responses and revised manuscript have adequately addressed your concerns, we would appreciate if you could consider revising your score.** Your support is invaluable to us and greatly motivates our ongoing efforts in this field.
>
> Thank you again for your time. Your constructive feedback has significantly enhanced the quality of our revised paper.
>
> Best,
>
> Paper 3307 Authors

---

> ### Author Response · Authors · 2024-12-03
> **Final Reminder for Reviewing Our Rebuttal**
>
> Dear Reviewer KmNX,
>
> We hope this message finds you well. As today is the last day for us to provide comments, we are reaching out with a final message to draw your attention to our rebuttal and revised manuscript. We have carefully addressed all the concerns initially raised in your reviews through **additional experiments, detailed clarifications, and revisions to our manuscript**. These updates directly respond to the key issues and suggestions highlighted in your initial assessments.
>
> Given the extensive efforts we have invested in addressing these points, we kindly encourage you to review our rebuttal and revised manuscript. We believe that our responses and additional experiments presented in our rebuttal effectively address the concerns, and we hope this will encourage you to revisit the score you previously assigned to our paper. **Your insights are incredibly important to us, and your reevaluation would be greatly appreciated.**
>
> Thank you for your attention and contribution to the review process.
>
> Warm regards,
>
> Authors of 3307

---

### Official Review · Reviewer_Sdkc · 2024-10-28

**Soundness:** 2
**Presentation:** 3
**Contribution:** 2
**Rating:** 5
**Confidence:** 5

**Summary:**

The paper introduces CLoRA, a novel approach for composing multiple Low-Rank Adaptation (LoRA) models to generate images that represent a variety of concepts in a single image. CLoRA addresses the challenges of attention overlap and attribute binding that arise when multiple LoRA models are used together. By revising the attention maps at test-time and using contrastive learning, CLoRA successfully creates composite images that accurately reflect the characteristics of each LoRA model. Also, the paper provides a comprehensive evaluation, demonstrating CLoRA's superiority over existing methods in multi-concept image generation using LoRAs.

**Strengths:**

1 CLoRA's ability to adjust attention maps at test-time allows for dynamic composition of multiple LoRA models without the need for retraining.
2 The use of contrastive learning to refine attention maps is a clever strategy that enhances the model's ability to generate images that respect the boundaries and characteristics of each LoRA model.
3 CLoRA directly addresses the issues of attention overlap and attribute binding, which are critical in the context of multi-LoRA composition.

**Weaknesses:**

1 The prompts used in the evaluation phase of the paper are relatively simple and do not fully explore the model's performance in handling complex scenes involving interactions between multiple objects. Therefore, the capability of CLoRA in generating composite images with complex interacting objects has not been sufficiently validated.
2 The use of contrastive learning in the paper aims to align the attention of objects introduced by LoRA with specific concepts. A possible question is why not directly generate masks based on the attention of these specific concepts, rather than adopting the method of contrastive learning?
3 CLoRA has a limitation in the number of LoRA objects it can support, with the current method only effectively handling combinations of up to four objects. This poses a limitation for creating richer or more complex scenes, especially in creative tasks that require combinations of more elements.

**Questions:**

None

---

> ### Author Response · Authors · 2024-11-23
>
> Thank you for the supportive feedback. We respond to your comments below. Please feel free to let us know if you have any further questions.
>
> **Prompts with complex scenes and interactions:** Thanks for pointing this out. While we used simple prompts in line with prior research and readability, our method *does not constrain the usage of complex scenes or interactions between objects*. CLoRA focuses mainly on managing the attention maps of subject tokens, ensuring their separation and fidelity while allowing other tokens to coexist and interact freely. Importantly, we *do not explicitly diminish the attention values of tokens* unrelated to the subject tokens, enabling the model to naturally handle contextual and interaction-based details in the scene.
> To illustrate our method’s effectiveness, we generated additional results with the following text prompts: *“An L1 cat and an L2 dog playing with a ball together, near the beach with a ship in the background”* and *“An L1 cat and an L2 dog eating from the same plate, in a playground”.*  Please see Fig. 11  in the revised manuscript. As can be seen from the figures, our method *can handle the interaction between objects* such as a cat and a dog playing with a ball or eating from a plate. Moreover, our submitted copy includes examples where two subjects interact; such as Fig. 5 (b) where *a person is holding a bunny*, and Fig 5 (c) where a *person is holding a flower* (both people, bunny, and flowers are depicted by different LoRA models). Therefore, *our model does not have a limitation in terms of handling more complex scenes.*
>
> **Why not directly generate masks based on attention rather than a contrastive approach:** Thank you for your insightful question. While generating masks based solely on the attention maps of specific concepts might seem sufficient at first glance, our ablation study (Section 4.2 and Figure 6) demonstrates that masking alone cannot fully resolve the challenges of attribute binding and attention overlap. Without the latent updates introduced by our contrastive framework, the model often struggles to correctly distinguish between multiple concepts. This limitation can lead to issues such as duplicate objects, incorrect attribute binding (e.g., generating two dogs instead of a cat and a dog), or the omission of certain concepts.
> Moreover, in cases of catastrophic neglect, where the cross-attention mechanism fails to allocate attention adequately to some subjects, the masks derived from these attention maps are inherently incomplete or inaccurate, as discussed by earlier works such as Attend & Excite. This failure results in subjects being omitted or misrepresented in the final image. The *latent updates provided by our contrastive framework mitigate this issue* by ensuring that each subject receives distinct and sufficient attention, enabling the model to faithfully incorporate all specified concepts into the composition. In contrast, masking alone, without latent updates, lacks this consistency and fails to prevent attribute blending or omissions, as evidenced by our experiments.
>
> **CLoRA has a limitation in the number of LoRA objects it can support:** Thanks for pointing this out. We would like to highlight that supporting compositions with a large number of LoRA models is a *challenging task for all competitor methods* as well. This is primarily due to the memory consumption of individual LoRA models and the difficulty in effectively separating multiple LoRA concepts without blending them. Upon further examination of competing methods, we observed that ZipLoRA showcased compositions up to 2 LoRA models, MultiLoRA up to 3 LoRAs, and Mix of Show up to 5 LoRA models. Also, as can be seen in Figure 5, while our method faithfully generates multiple concepts such as 3 successfully, other methods such as LoRA Merge and MultiLoRA fail to generate the correct concepts. It is important to note, however, that Mix of Show requires additional inputs from the user, such as human pose or canny edges, whereas our method operates without these conditions.
> Further experiments conducted on a larger GPU (80 GB) allowed our model to handle up to 8 LoRA compositions, with runtimes ranging from 24 seconds (for 2 LoRAs) to 95 seconds (for 8 LoRAs).  *We have included a discussion and analysis of these findings in Appendix A, Table 2, and Figure 7 of the revised manuscript.*

---

> > ### Author Response · Authors · 2024-12-01
> >
> > Dear Reviewer Sdkc,
> >
> > We would like to kindly remind you that tomorrow is the deadline for reviewers to submit their comments. Should you have any further questions or need additional clarification, please do not hesitate to reach out.
> >
> > We addressed all your concerns (see our rebuttal response above) and revised our manuscript accordingly. **If you find our responses and revised manuscript have adequately addressed your concerns, we would appreciate if you could consider revising your score.** Your support is invaluable to us and greatly motivates our ongoing efforts in this field.
> >
> > Thank you again for your time. Your constructive feedback has significantly enhanced the quality of our revised paper.
> >
> > Best,
> >
> > Paper 3307 Authors

---

> ### Author Response · Authors · 2024-12-03
> **Final Reminder for Reviewing Our Rebuttal**
>
> Dear Reviewer Sdkc,
>
> We hope this message finds you well. As today is the last day for us to provide comments, we are reaching out with a final message to draw your attention to our rebuttal and revised manuscript. We have carefully addressed all the concerns initially raised in your reviews through **additional experiments, detailed clarifications, and revisions to our manuscript**. These updates directly respond to the key issues and suggestions highlighted in your initial assessments.
>
> Given the extensive efforts we have invested in addressing these points, we kindly encourage you to review our rebuttal and revised manuscript. We believe that our responses and additional experiments presented in our rebuttal effectively address the concerns, and we hope this will encourage you to revisit the score you previously assigned to our paper. **Your insights are incredibly important to us, and your reevaluation would be greatly appreciated.**
>
> Thank you for your attention and contribution to the review process.
>
> Warm regards,
>
> Authors of 3307

---

### Official Review · Reviewer_qREh · 2024-11-03

**Soundness:** 2
**Presentation:** 2
**Contribution:** 2
**Rating:** 5
**Confidence:** 5

**Summary:**

This paper presents a contrastive approach to seamlessly integrate multiple content and style LoRAs simultaneously, which works in test-time and does not require training. It effectively solves the issues of attention overlap and attribute binding.

**Strengths:**

1. The proposed method is simple and easy-to-follow.
2. The experimental results verify the effectiveness of the method, which surpasses the baseline methods.

**Weaknesses:**

1. The issue of attribute binding has been proposed in many works in the field of multi-concept image customization, such as [1] and [2]. The observation mentioned in this paper is not novel.
2. There has been some works on designing attention mechanisms to address attention overlap, such as [2]. It needs additional experiments to verify the advantages of the proposed contrastive way,
3. Test-time optimization introduces additional inference time, which requires experimental comparison.
4. The dataset in the proposed benchmark only has 20 more characters than CustomConcept101, which lacks a sufficient contribution.
5. The evaluation metric used in quantitative experiments are not enough. Clip-T and Clip-I should also be compared.

[1] Attend-and-excite: Attention-based semantic guidance for text-to-image diffusion models. ACM Transactions on Graphics (TOG), 42(4):1–10, 2023.

[2] Attention Calibration for Disentangled Text-to-Image Personalization. CVPR 2024.

**Questions:**

As seen in weaknesses.

---

> ### Author Response · Authors · 2024-11-23
>
> Thank you for the supportive feedback. We respond to your comments below in 2 seperate answers due to length limitations. Please feel free to let us know if you have any further questions.
>
> **Novelty of Observations Mentioned in the Paper:** We appreciate the reviewer pointing out prior works on attribute binding and attention overlap, such as Attend-and-Excite [1] and DisenDiff [2]. As noted in our paper (Lines 097–099), Attend-and-Excite [1] as well as other related works have highlighted these challenges in image generation. Our novel observation, presented for the first time in this paper, is that these issues also arise in LoRA composition tasks. Specifically, when multiple LoRA models are composed, their attentions can overlap, leading to failures in generating the intended compositions. While this observation is novel in itself, the main contribution and the core technical innovation of our work lies in the proposed contrastive objective function.  Unlike existing methods that focus on a single model or require specialized variants (e.g., EDLoRAs), or user-provided controls, our approach operates on community-standard LoRAs in a plug-and-play, training-free, and test-time manner. This enables the seamless combination of LoRAs without blending or losing fidelity, as shown in our results (Figures 3–5, Table 1). *To the best of our knowledge, both the observation regarding attention overlap in LoRA composition and the design of our method are novel contributions.*
>
> **Additional Experiments with DisenDiff[2]:** DisenDiff focuses on disentangling multiple concepts within a given image using attention calibration techniques. In contrast, our method addresses a fundamentally different problem: the composition of multiple pre-trained LoRA models into a single image. Our work deals with managing multi-LoRA attention overlap and attribute binding across independently trained LoRA models, enabling seamless and faithful LoRA compositions.  Note that DisenDiff is not capable of creating images using multiple LoRA models unless a composed image is given as an input in the first place. For instance, if one wants to create a composition using a cat LoRA and a dog LoRA, DisenDiff would require an image that already depicts both concepts, to begin with. Composing both concepts in a single image is exactly the problem we are interested in. Our method leverages community-developed LoRA models to compose entirely new multi-concept images. This difference highlights that DisenDiff is not a framework for concept merging but rather one for disentangling concepts already present in a given image, making direct comparisons between the two approaches not possible. However, *we included this work in our Related Work* section in the updated manuscript, please see Section 2.
>
> **Additional Time for Test-time Optimization:** Our method only requires test-time optimization, resulting in no additional time consumption beyond this phase. Unlike many of the competitors, we do not require finetuning or training. This test-time optimization *takes 24 seconds* for image composition using 2 LoRA models in H100 GPUs. Additionally, we have included *a detailed comparison of runtime between our method and competing approaches*, as detailed below. Also, we revised our manuscript to include a detailed discussion, please *refer to Appendix A.*
>
> | Method                | CivitAI Compatibility | VRAM (Finetuning/Inference) | Runtime (Finetuning/Inference) |
> |-----------------------|-----------------------|-----------------------------|---------------------------------|
> | Custom Diffusion      | no                   | 28GB + 8 GB                 | 4.2 min + 3.5s                 |
> | LoRA Merge            | yes                  | 7 GB                        | 3.2 s                          |
> | Multi-LoRA - composite| yes                  | 7 GB                        | 3.4 s                          |
> | Multi-LoRA - switch   | yes                  | 7 GB                        | 4.8 s                          |
> | Mix-of-Show           | no                   | 10GB + 10 GB                | 10min + 3.3s                   |
> | ZipLoRA               | yes                  | 39GB + 17 GB                | 8min + 4.2s                    |
> | OMG                   | yes                  | 30 GB                       | 62 s                           |
> | LORA Composer         | no                   | 51 GB                       | 35 s                           |
> | Ours                  | yes                  | 25 GB                       | 24 s                           |
>
>
> **Please see our next answer for other comments.**

---

> ### Author Response · Authors · 2024-11-23
>
> **Insufficient contribution with the proposed dataset:** As detailed in our Appendix, our benchmark dataset builds upon CustomConcept101. However, we would like to note that our contribution extends beyond merely adding 20 new concepts to CustomConcept101. While CustomConcept101 features multiple images across 101 concepts, it only includes 15 LoRA models and provides 125 unique prompts for composition. In contrast, we have trained 131 LoRA models that encompass both CustomConcept101 and our additional concepts. Furthermore, we offer 200 prompts designed for multi-LoRA composition, ranging from simple to complex arrangements. The *contribution of our benchmark dataset lies in these trained LoRA models and the expanded set of prompts*, all available for public use. However, we *revised our manuscript to clarify these contributions*, and instead of referring to it as a ‘benchmark dataset’ we changed it to *‘benchmark LoRA collection’* throughout the paper’. Please refer to the revisions in our manuscript, specifically Lines 895-897, 912-915, and Appendix D.
>
>
> **Clip-T and Clip-I should also be compared:** Thank you for this suggestion. Based on your request, we computed CLIP-I and CLIP-T scores, as can be seen from the following table. Note that our method performed the *highest scores* for both CLIP-T and CLIP-I scores. These results also support the DINO metrics presented in the main paper. We revised our manuscript to include these metrics, along with a discussion. *Please see our revised manuscript, Table 1.*
>
>
> |                |               | Merge [Ryu et al., 2023] | Composite | Switch [Zhong et al., 2024] | ZipLoRA [Shah et al., 2023] | Mix-of-Show [Gu et al., 2023] | **Ours**         |
> |----------------|---------------|--------------------------|-----------|-----------------------------|-----------------------------|-------------------------------|------------------|
> | **CLIP-I**     | **Min.**      | 0.641 ± 0.029           | 0.614 ± 0.035 | 0.619 ± 0.039           | 0.659 ± 0.022           | 0.664 ± 0.023               | **0.683 ± 0.017** |
> |                | **Avg.**      | 0.683 ± 0.029           | 0.654 ± 0.035 | 0.659 ± 0.036           | 0.707 ± 0.021           | 0.712 ± 0.022               | **0.725 ± 0.017** |
> |                | **Max.**      | 0.714 ± 0.028           | 0.690 ± 0.033 | 0.695 ± 0.036           | 0.740 ± 0.021           | 0.744 ± 0.023               | **0.756 ± 0.017** |
> | **CLIP-T**     |               | 0.814 ± 0.054           | 0.833 ± 0.091 | 0.822 ± 0.089           | 0.767 ± 0.081           | 0.760 ± 0.074               | **0.862 ± 0.052** |

---

> ### Author Response · Authors · 2024-12-01
>
> Dear Reviewer qREh,
>
> We would like to kindly remind you that tomorrow is the deadline for reviewers to submit their comments. Should you have any further questions or need additional clarification, please do not hesitate to reach out.
>
> We addressed all your concerns (see our rebuttal response above) and revised our manuscript accordingly. **If you find our responses and revised manuscript have adequately addressed your concerns, we would appreciate if you could consider revising your score.** Your support is invaluable to us and greatly motivates our ongoing efforts in this field.
>
> Thank you again for your time. Your constructive feedback has significantly enhanced the quality of our revised paper.
>
> Best,
>
> Paper 3307 Authors

---

> > ### Comment · Reviewer_qREh · 2024-12-02
> >
> > Thanks for the rebuttal, which addresses some of my concerns. However, my primary concern lies in the novelty of the observations, as the combination of LoRA is merely a specific instance of multi-concept composition and prior research has identified attention overlap and attribute binding issues for multi-concept composition. Hence, I maintain the original score.

---

> > > ### Author Response · Authors · 2024-12-02
> > >
> > > We appreciate your feedback on our rebuttal. We would like to address your comments to clarify any confusion, regarding  "combination of LoRA being merely a specific instance of multi-concept composition" and "prior research identifying attention overlap and attribute binding issues for multi-concept composition". **We believe these points may reflect a misunderstanding, which we aim to clarify below.**
> > >
> > > The "multi-concept composition" we discuss **differs fundamentally from existing work that operates within single-model image generation**. In these models, "multi-concept" refers to **token-level "concepts"** (e.g., concepts being "cat" or "dog" tokens). In contrast, in our case, "multi-concepts" refer to **individual LoRA models**, each representing a different fine-tuned Stable Diffusion model for a specific concept. While attribute binding in single-model image generation ties to **token overlaps** (e.g., overlapping attention of cat and dog tokens), multi-LoRA generation concerns about multiple latent representations that introduces **fundamentally different and more complex challenges as detailed below.**
> > >
> > > In single-model compositions, the specific spatial location of objects like a cat or a dog is less critical, as long as their attention maps are adequately separated to prevent attention overlap and attribute binding, thereby addressing issues of **"semantic" confusion.**  Ensuring that each object's features remain distinct is sufficient to generate coherent outputs. However, in our multi-LoRA context, the challenge is significantly more complex. In addition to managing semantic confusion, **we must also ensure spatial consistency across multiple independently trained LoRA models**, each contributing to the composition with its latent representation. If the spatial locations of objects across these latents are not aligned properly, the resulting composition risks being affected by binding or overlap issues.
> > >
> > > For example, consider a composition involving two LoRA models, where L1 represents a cat and L2 represents a dog. In the L1 LoRA model, the cat might naturally appear on the left side of the image and the dog on the right. In contrast, the L2 LoRA model might place the cat on the right and the dog on the left due to differences in their independently trained features and attention maps. **If these latents are simply fused**, as in methods like MultiLoRA-Composite, significant issues arise: one concept might dominate (e.g., the output shows two cats), or features from different objects could improperly blend (e.g., a cat and a dog appear, but the dog adopts the cat's color palette or texture). These problems stem from misaligned attention maps and overlapping contributions, leading to attribute binding and conceptual inconsistencies. **This challenge is fundamentally different from the token-level concerns addressed in works like Attend & Excite**, which focus on managing attention within a single model. Our observation, in contrast, specifically shows the complexities of **multi-model compositions, where both spatial and semantic alignment** are required to address due to multiple latent representations (a situation not faced by single-models).
> > >
> > > **Therefore, the combination of LoRAs is not merely a simple instance of the compositions tackled in works like Attend & Excite.** To the best of our knowledge, this is the **first work that demonstrates this additional level of spatial confusion across multiple LoRA models** (as opposed to token-level bindings or overlaps). However, we kindly note that this observation is **not our main technical contribution.** As stated in our rebuttal, **our technical contribution is a novel objective function that separates the attentions of multiple LoRA models in test-time.** Compared to other methods, our method is able to generate successful compositions (Figures 3–5, Table 1) while other methods either require specialized variants (e.g., EDLoRAs), or user-provided controls, which can limit their practicality. Moreover, **our method can scale up to 8 LoRAs** (while other methods demonstrated up to 5 LoRA models), taking from **25 seconds (2 LoRAs)  to 95 seconds (8 LoRAs)** without losing fidelity.
> > >
> > > However, we acknowledge your feedback, and **we are committed to revising our manuscript to avoid any confusion.** To prevent readers from confusing "multi-concept" composition with single-model compositions, **we will incorporate the above discussion into our final manuscript**, ensuring that our observation is well articulated, and also the **focus placed clearly on our primary technical achievement.** We appreciate your thoughtful feedback, which is instrumental in improving the clarity and impact of our work.
> > >
> > > **We kindly ask you to reconsider your evaluation based on this clarification, and on the novelty and effectiveness of our technical contribution**, particularly in how it addresses the limitations of prior approaches and advances the state of the art.

---

### Official Review · Reviewer_4FCc · 2024-11-04

**Soundness:** 3
**Presentation:** 3
**Contribution:** 2
**Rating:** 6
**Confidence:** 4

**Summary:**

This paper introduces a new training-free approach to utilizing multiple LoRA models to generate images with high quality and distinctive features, which solving the attention overlap problem and attribute binding problem by updating the attention maps of multiple LoRA models at test-time. The result shows improved performance in both qualitative assessment and quantitative assessment.

**Strengths:**

1. This paper is detailed in the background introduction. Specific and vivid examples are used to illustrate the pitfall of previous work and the advantage of cLoRA. It explains the root causes of existing work defects and get nice result using new model.
2. A contrastive objective-based approach to integrate multiple content and style LoRAs simultaneously without training.
3. A method that can use community LoRAs in a plug-and-play manner without needing specialized variants.

**Weaknesses:**

1. In user study of Section 4, it says the user study involves 50 participants, but does not clarify how many samples each participant was shown. Also, this paper seems to be too brief in the limitation and conclusion sections. For example, in Section 5 when mentioning processing times and the number of LoRA models, the data is not showed to better illustrate.
2. The core innovation of the method is to differentiate between different concepts using distinct attention maps. However, this design lacks novelty, as similar approaches have been employed in previous research. Additionally, while attention maps are primarily used to control the separation of different concepts, they also restrict information exchange between these concepts. This phenomenon is evident in the results presented.
3. The inference phase consumes more computational resources and the computational complexity increases linearly with the number of merged LoRA models

**Questions:**

None

**Details Of Ethics Concerns:**

The paper uses images of certain public figures, and the authors need to demonstrate that such use does not involve copyright infringement.

---

> ### Author Response · Authors · 2024-11-23
>
> Thank you for the supportive feedback. We respond to your comments below. Please feel free to let us know if you have any further questions.
>
> **User study details:** The user study was made on 50 participants, where each participant was shown 48 images. The order of images shown to the users was randomized. *We revised our manuscript to add screenshots (Please refer to Figure 9) and more details* about the user study (please refer to the revised manuscript, Appendix B).
>
> **Too brief limitation and conclusion sections/data for processing times and number of LoRA models:** Thanks for pointing this out. Based on your suggestion, *we added a figure* showing the processing time vs. the number of LoRA models for up to 8 LoRA models (see Fig. 7 in the revised manuscript). We also *extended the discussion* in both the limitation and conclusion sections (please refer to Section 5 and Section 6 in the revised manuscript).  Also, for a more in-depth exploration of the runtime and resource consumption of our method and competitors, please see Appendix A in our revised manuscript.
>
> **Core innovation lacks novelty:** As highlighted in our paper (Lines 097-099), prior research has identified attention overlap and attribute binding issues as challenges in image generation tasks. *Our novel observation*, presented for the first time in this paper, is that these issues also arise in LoRA composition tasks. Specifically, when multiple LoRA models are composed, their attentions can overlap, leading to failures in generating the intended compositions. While this observation is novel, *the core technical innovation* of our work lies in the proposed contrastive objective function. This objective effectively separates the attention of multiple LoRA models during test-time, without requiring any additional training, user-provided masking, or conditions. *To the best of our knowledge, both the observation regarding attention overlap in LoRA composition and the design of our method are novel contributions.* Moreover, both our visual and quantitative results (Figures 3–5, Table 1) demonstrate superior performance for handling LoRA compositions compared to competing methods, showing the efficiency of the proposed method.
>
> **Attention maps restrict information exchange between concepts:** We would like to clarify this misunderstanding. The contrastive loss we designed in this paper does not enforce physical separation or distance between subjects in the generated image. Instead, it encourages that the attention maps corresponding to different subject-specific tokens (such as cat or dog) are non-overlapping. *We note that this does not restrict information exchange or interactions between subjects.* This is because the attention to tokens that are responsible for contextual relationships, spatial interactions, or broader scene coherence remains unconstrained and, therefore tokens can coexist and interact with each other. Additionally, we do not restrict self-attention, which ensures that pixels can interact freely with one another, preserving the flexibility and coherence of the diffusion process. Given that our contrastive objective operates in an optimization-based manner, based on a given text prompt, the degree of separation between attention maps across subjects will be optimized in a way that minimizes attention overlap and attribute binding; while still allowing for the generation of the context described by the specified text prompt.  Our results (e.g., Figures 3–5) demonstrate that this balance enables the accurate and natural composition of multiple LoRA concepts without sacrificing the interplay between elements in the image.
>
> **Ethical Concerns:** Thank you for raising this point. We initially used celebrity images in our work since related papers such as [OMG (ECCV’24)](https://kongzhecn.github.io/omg-project) have extensively utilized celebrity images. Additionally, the use of well-known faces allows readers to easily assess whether facial preservation has been maintained effectively. However, upon further reflection and research, we acknowledge that using these images without the celebrities’ consent could be considered unethical. For the camera-ready version of our paper, we will replace the celebrity images with the authors' images. Unfortunately, due to anonymity constraints, we were unable to make this change in the current submission. Thank you again for bringing this to our attention, and *we will remove celebrity images from our camera-ready version.*

---

> ### Author Response · Authors · 2024-12-01
>
> Dear Reviewer 4FCc,
>
> We would like to kindly remind you that tomorrow is the deadline for reviewers to submit their comments. Should you have any further questions or need additional clarification, please do not hesitate to reach out.
>
> We addressed all your concerns (see our rebuttal response above) and revised our manuscript accordingly. **If you find our responses and revised manuscript have adequately addressed your concerns, we would appreciate if you could consider revising your score.** Your support is invaluable to us and greatly motivates our ongoing efforts in this field.
>
> Thank you again for your time. Your constructive feedback has significantly enhanced the quality of our revised paper.
>
> Best,
>
> Paper 3307 Authors

---

> ### Author Response · Authors · 2024-12-03
> **Final Reminder for Reviewing Our Rebuttal**
>
> Dear Reviewer 4FCc,
>
> We hope this message finds you well. As today is the last day for us to provide comments, we are reaching out with a final message to draw your attention to our rebuttal and revised manuscript. We have carefully addressed all the concerns initially raised in your reviews through **additional experiments, detailed clarifications, and revisions to our manuscript**. These updates directly respond to the key issues and suggestions highlighted in your initial assessments.
>
> Given the extensive efforts we have invested in addressing these points, we kindly encourage you to review our rebuttal and revised manuscript. We believe that our responses and additional experiments presented in our rebuttal effectively address the concerns, and we hope this will encourage you to revisit the score you previously assigned to our paper. **Your insights are incredibly important to us, and your reevaluation would be greatly appreciated.**
>
> Thank you for your attention and contribution to the review process.
>
> Warm regards,
> Authors of 3307

---

### Author Response · Authors · 2024-11-23

**Dear Reviewers and ACs,**

Thank you for dedicating time and effort to review our paper! Your insightful comments have been invaluable in refining our work, and we have addressed the concerns point by point.

We have taken all your suggestions carefully and updated our previous version accordingly. According to our understanding, there are two common concerns from reviewers that we thoroughly addressed both in our answers and in the revised manuscript: runtime and scalability analysis; and clarification of the novelty of our method.

**In the revised manuscript, we made the following changes, which are highlighted in green in the updated document:**

1) Added a runtime comparison with competing methods, please see Appendix A.1 and Table 2.
2) Added a scalability analysis as we increase the number of LoRAs, please see Appendix A.2. and Fig.7.
3) Added more details about our user study, please see Appendix B.
4) Added requested metrics CLIP-I and CLIP-T for comparison, please see updated Table 1.
5) Added more qualitative results with the same type of subjects (e.g. two cats) and more complex scenes and prompts, please see Appendix C and Fig 10, and Fig 11.
6) Added clarification about the contribution of the benchmark dataset, please see Appendix D.
7) Extended Limitations and Conclusion sections, please see Section 5 and 6.
8) Added requested related works, please see Section 2.
9) Fixed typos and notations, please refer to individual answers or highlighted sections throughout the revised paper.
10) Added qualitative comparisons with LoRA-Composer, please see Appendix E.

**We are happy to discuss with you further if you still have any additional concerns.** Thank you once again!

Best regards,

Paper 3307 Authors

---

> ### Author Response · Authors · 2024-11-29
>
> Dear Reviewers,
>
> Since the rebuttal deadline is ending soon, we wanted to send a kind reminder about our responses and revised manuscript. We addressed all your concerns point by point, and revised our manuscript to reflect your suggestions. Your feedback is very important to us, and we will do our best to answer any further questions or concerns as soon as possible.
>
> Best,
>
> Paper 3307 Authors

---

### Meta-Review · Area_Chair_B8RC · 2024-12-17

**Metareview:**

This paper presents an image generation algorithm to integrate multiple concepts using multiple LoRA models. The key is to use a contrastive learning framework to guide the attention of these LoRA models, preventing them from conflicting with each other. The proposed method makes sense and the generated images look good, but as (almost all) the reviewers pointed out, the novelty is limited given the existing literature on image generation algorithms guided by attributes or instances. After the rebuttal, most reviewers are still slightly negative about the submission. The AC decides to follow the reviewers' overall suggestion and recommends rejection. The authors are suggested to make the technical contribution clearer, in particular, highlighting the difference from existing algorithms working on similar scenarios.

**Additional Comments On Reviewer Discussion:**

The paper initially got a rating of 5/5/5/5/6. After the rebuttal, only one negative reviewer raised the score from 5 to 6, which is yet insufficient to make the paper accepted.

---

### Decision · Program_Chairs · 2025-01-22

Reject